# SoftSignSGD(S3): An Enhanced Optimizer for Practical DNN Training and Loss Spikes Minimization Beyond Adam

## Abstract

Adam has been widely successful in training deep neural networks (DNNs), yet the factors contributing to both its practical effectiveness and ineffectiveness remain largely underexplored. In this study, we reveal that the effectiveness of Adam in training complicated DNNs stems primarily from its similarity to SignSGD in managing significant gradient variations, while we also theoretically and empirically uncover that Adam is susceptible to loss spikes due to potential excessively large updates. Building on these insights, we propose a novel optimizer, SignSoft-SGD (S3), which incorporates a generalized sign-like formulation with a flexible $p$-th order ($p \geq 1$) momentum in the denominator of the update, replacing the fixed 2-order momentum. We also integrate the memory-efficient Nesterov's accelerated gradient technique into S3, enhancing convergence speed without additional memory overhead. To minimize the risk of loss spikes, we utilize the same coefficient for the momentums in both the numerator and the denominator of the update, which also practically streamlines the tuning overhead. We conduct a theoretical analysis of S3 on a general nonconvex stochastic problem, demonstrating that S3 achieves the optimal convergence rate under weak assumptions. Extensive experimentation across various vision and language tasks demonstrates that S3 not only achieves rapid convergence and improved performance but also rarely encounters loss spikes even at a $10\times$ larger learning rate. Specifically, S3 delivers performance comparable to or better than AdamW with $2\times$ the training steps.

## 1 Introduction

Optimizers play a pivotal role in training DNNs.Currently, Adam (Kingma & Ba (2015)) stands out as the dominant optimizer for training Transformers (Vaswani et al. (2017)), especially for the recent phenomenal large language models (LLMs) (Brown et al. (2020); Chowdhery et al. (2023); Touvron et al. (2023)), and large vision models (Radford et al. (2021); Kirillov et al. (2023)). Even in the realm of training the modern convolutional neural networks (CNNs), such as ConvNeXt (Liu et al. (2022); Woo et al. (2023)), Adam also has become the de facto optimizer, although stochastic gradient descent (SGD) was traditionally deemed more suitable for training CNNs (Krizhevsky et al. (2017); He et al. (2016)).

While the practical success of Adam is indisputable, the underlying reasons for its effectiveness remain poorly understood. The original paper on Adam attributes its success to the effective coordinate-wise adaptivity (Kingma & Ba (2015)). However, recent work (Chen et al. (2023b)) challenges this perspective by proposing an optimizer that achieves comparable, and sometimes superior, performance to Adam in training various architectures without leveraging adaptivity.

We first revisit Adam to discern the reasons behind its practical effectiveness. Each coordinate of the update of Adam (*i.e.*, $\frac{m_t}{\sqrt{v_t}}$) exhibits a sign-like characteristic. Empirical evidence from (Kunstner et al. (2023)) demonstrates that the simple sign descent can substantially narrow the performance gap between SGD and Adam in training complicated DNNs, such as Transformers. *This suggests that the sign-like property of Adam is a key factor in its effectiveness.* However, (Kunstner et al. (2023)) uncovered this phenomenon but did not explore why sign descent is effective in training Transformers. This paper explains that wide difference in inter-layer and intra-layer gradients during training is the behind reason, and the effectiveness of Adam is mainly attributed to its conservative

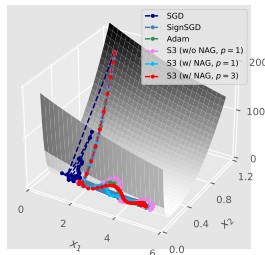 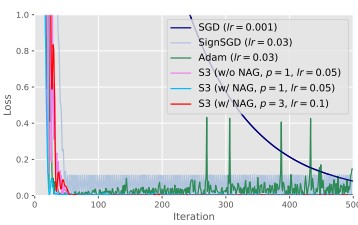 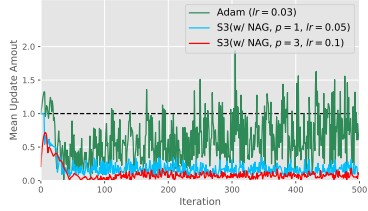

(a) Trajectories in 3D space     (b) Dynamics of Loss Convergence     (c) Dynamics of Mean update

Figure 1: The (a) trajectories, (b) loss convergence curves, and (c) mean update curves of SGD, SignSGD, Adam and S3. The loss function is defined as $f(x^{(1)}, x^{(2)}) = 0.5(x^{(1)} - 1/x^{(2)})^2 + 0.5(x^{(1)} - 20x^{(2)})^2$. The initial point is set as $[x_0^{(1)}, x_0^{(2)}] = [1.0, 1.0]$. Gaussian noise is added to the gradient at each iteration to simulate random sampling, represented as $g_t = [\nabla f_{x_t^{(1)}}, \nabla f_{x_t^{(2)}}] + \mathcal{N}(0, 0.1)$. Due to the significant gap between $\nabla f_{x_t^{(1)}}$ and $\nabla f_{x_t^{(2)}}$, we set a small learning rate for SGD to prevent divergence. However, this results in slow convergence. The update of SignSGD, $\text{Sign}(g_t)$, causes the loss to oscillate and prevents it from converging to the minimum. The sign-like property of Adam makes it perform much better than SGD. The update of Adam, $\frac{m_t}{\sqrt{v_t}}$, is generally smaller, aiding in achieving a lower loss, compared to SignSGD. However, it has a non-trivial probability of encountering loss spikes. The update of S3 is constrained to $[-1, 1]$ and gradually diminishes as the loss approaches the minimum. This property enables S3 to achieve an extremely small loss and seldom encounter loss spikes. The introduction of the NAG technique in S3 is helpful for accelerating convergence. The use of a large $p$-th order momentum allows S3 to employ a large learning rate without encountering training instability.

sign-like descent to address the problem of the great gradient discrepancy, when training complicated DNNs. However, we demonstrate that Adam is also the main contributing factor to the risk of training instability and loss spikes. This is because each coordinate of the update of Adam potentially reach excessively values with non-trivial probability.

Drawing insights from the analyses of Adam, we propose a novel optimizer, called SignSoftSGD (S3). *First*, S3 features a more generalized sign-like formulation with a flexible $p$-th order ($p \geq 1$) momentum in the denominator of the update, rather than being limited to a fixed 2-order momentum like Adam. This modification enables S3 to utilize a larger $p$-order momentum, allowing for a larger learning rate for faster convergence and improved performance without the risk of training instability, as observed with Adam. *Second*, S3 employs the same exponential moving average coefficient $\beta$ for both the numerator and the denominator momentums in the update, strictly constraining the update within the range of $[-1, 1]$[1]. Consequently, S3 seldom encounters loss spikes, thanks to its strategy of minimizing the maximum update. Additionally, it offers practical advantages by eliminating the need for bias correction and gradient clipping, while reducing tuning efforts due to one less hyperparameter. *Third*, we introduce the technique of an equivalent Nesterov's accelerated gradient (NAG) to S3, further enhancing training speed without incurring additional memory costs. We provide an illustrative example visualizing the convergence behaviors of SGD, SignSGD, Adam, and S3 in Figure 1. Furthermore, we offer theoretical convergence rate analysis for S3 on a general nonconvex stochastic problem, aligning with the theoretical lower bound of the optimal convergence rate $O(\frac{1}{T^{1/4}})$, where $T$ represents the number of iterations under weak assumptions.

Our primary contributions are summarized in the following:

- **We theoretically and empirically demonstrate that Adam is the underlying factor causing loss spikes in training large models (*i.e.*, LLM)** due to its potential to lead some parameter updates to be excessively large. (Section 2)

- **We introduce a novel optimizer, named S3, which offers four distinct advantages over Adam**: 1) elimination of bias correction as well as gradient clipping, reducing one hyperparameter, 2) a generalized formulation enabling the utilization of larger learning rates for improved performance, 3) integration of an equivalent NAG technique to accelerate training convergence without additional memory overhead, and 4) minimized risk of training instability and loss spikes. (Section 3)

---

[1]This is the reason why we named the new optimizer SoftSignSGD.

- We conduct a theoretical analysis for S3 on a general nonconvex stochastic problem, **achieving the optimal convergence rate under a weak non-uniform smoothness assumption.** (Section 4)
- We conduct extensive experiments to evaluate S3 against Adam and other related optimizers. The experimental results demonstrate the faster training speed and superior inference performance of S3. Specifically, **Specifically, S3 achieves improvements comparable to or exceeding those of AdamW with twice the training steps, while rarely experiencing loss spikes even at significantly higher learning rates.** (Section 5)

## 2 RETHINKING THE EFFECTIVENESS AND INEFFECTIVENESS OF ADAM

In a deep learning task, the optimizer aims to minimize the empirical risk loss of a model on a dataset, *i.e.*,

$$\min_{x \in \mathbb{R}^d} F(\boldsymbol{x}) = \mathbb{E}_{\zeta \sim \mathcal{D}}[f(\boldsymbol{x}; \zeta)] = \frac{1}{n} \sum_{i=1}^{n} f(\boldsymbol{x}; \omega_i), \tag{1}$$

where $\boldsymbol{x}$ is the $d$-dimensional model parameter, and $\zeta$ is independently and identically sampled from the dataset $\{\omega_i : \omega_i \in \mathcal{D}, 1 \le i \le n\}$.

Nowadays, Adam has emerged as the dominant optimizer for training DNNs. It significantly outperforms SGD in training the increasingly popular Transformers, demonstrating remarkable efficacy. Even for CNN-based models like ConvNeXT, Adam is preferred for achieving superior performance, despite the historical consideration that SGD is more suitable for training CNNs. While the practical success of Adam is indisputable, the factors contributing to its practical effectiveness remain largely underexplored. There is a pressing need to delve into the effectivness of Adam to facilitate significant advancements in DNN training.

Recalling the updating rule of Adam, we have

$$\begin{aligned}
\tilde{\boldsymbol{m}}_t &= \beta_1 \tilde{\boldsymbol{m}}_{t-1} + (1 - \beta_1)\boldsymbol{g}_t, \\
\boldsymbol{m}_t &= \frac{\tilde{\boldsymbol{m}}_t}{1 - \beta_1^t}, \\
\tilde{\boldsymbol{v}}_t &= \beta_2 \tilde{\boldsymbol{v}}_{t-1} + (1 - \beta_2)\boldsymbol{g}_t^2, \\
\boldsymbol{v}_t &= \frac{\tilde{\boldsymbol{v}}_t}{1 - \beta_2^t}, \\
\boldsymbol{x}_{t+1} &= \boldsymbol{x}_t - \gamma_t \frac{\boldsymbol{m}_t}{\sqrt{\boldsymbol{v}_t}},
\end{aligned} \tag{2}$$

where $x_t$ denotes the model parameter, $\boldsymbol{g}_t = \nabla f(\boldsymbol{x}_t; \zeta_t)$ is the stochastic gradient, $\gamma_t$ is the learning rate, and $\beta_1$ and $\beta_2$ represents the exponential moving average coefficients.

In essence, $|\boldsymbol{m}_t|$ and $\sqrt{\boldsymbol{v}_t}$ are of the same order of magnitude. Specifically, if $g_t$ ideally stays stable over a period, Adam in Eq. (2) reduces to SignSGD, *i.e.*. $\boldsymbol{x}_{t+1} = \boldsymbol{x}_t - \gamma_t \frac{\boldsymbol{m}_t}{\sqrt{\boldsymbol{v}_t}} = \boldsymbol{x}_t - \gamma_t \text{Sign}(\boldsymbol{g}_t)$. Therefore, Adam can be viewed as a sign-like optimizer.

The primary reason for Adam's practical effectiveness over SGD in training complex DNNs remains fragmented across prior studies and lacks consolidation. Kunstner et al. (2023) empirically shows that sign descent with momentum yields comparable performance to Adam when training Transformers, albeit lacking comprehensive analytical justification. More recently, Chen et al. (2023b) employs an AutoML method to discover an effective optimizer, Lion, resembling SignSGD with momentum, and showcases superior performance to Adam across diverse DNN models. Indeed, the effectiveness of both Adam and Lion primarily stems from their shared sign-like property. For deep networks, the gradients of the initial and final layers can differ significantly, as theoretically verified in (Yang et al. (2019); Liu et al. (2020); Xiong et al. (2020); Kim et al. (2021); Qi et al. (2023)) through the mean-field theory and from the perspective of Lipschitz continuity. Furthermore, even within the same layer of a Transformer, gradients can vary greatly due to the presence of the attention component (Noci et al. (2022)). An illustrative example can be found in Section B.4 of the appendix. This substantial gradient discrepancy poses challenges for SGD, which, by directly employing gradients as updates, necessitates a relatively small learning rate to prevent divergence, resulting in noticeable training slowdown. Moreover, another drawback of using SGD is that parameters with large gradients undergo substantial changes, while those with small gradients tend

to remain close to their initial values. This discrepancy weakens the overall representation ability of networks, thereby degrading final performance. In contrast, Adam's updates remain close to $\pm1$ despite significant gradient gaps, thanks to its inherent sign-like property. This characteristic renders Adam a conservative yet effective choice for training complex DNNs. **In summary, Adam's efficacy in training complex DNNs stems from its conservative sign-like descent, which effectively addresses significant gradient discrepancies.**

While Adam effectively trains complex DNNs, it also escalates the risk of training instability and loss spikes with non-trivial probability. This can be inferred from Theorem 1.

**Theorem 1.** *The sequences $\{\boldsymbol{m}_t\}$ and $\{\boldsymbol{v}_t\}$ are generated by Adam in Eq. (2). If the moving average coefficients satisfy $\beta_1^2 < \beta_2$, then it holds that*

$$\frac{|\boldsymbol{m}_t^{(j)}|}{\sqrt{\boldsymbol{v}_t^{(j)}}} \leq \frac{(1-\beta_1)\sqrt{1-\beta_2^t}\sqrt{1-(\frac{\beta_1^2}{\beta_2})^t}}{(1-\beta_1^t)\sqrt{1-\beta_2}\sqrt{1-\frac{\beta_1^2}{\beta_2}}} \simeq \frac{1-\beta_1}{\sqrt{1-\beta_2}\sqrt{1-\frac{\beta_1^2}{\beta_2}}}, \tag{3}$$

*where $\frac{|\boldsymbol{m}_t^{(j)}|}{\sqrt{\boldsymbol{v}_t^{(j)}}}$ reach to the largest value if the signs of $\{\boldsymbol{g}_t^{(j)}, \boldsymbol{g}_{t-1}^{(j)}, ... \boldsymbol{g}_{t-k}^{(j)}...\}$ are the same and $|\boldsymbol{g}_t^{(j)}| = \frac{\beta_2|\boldsymbol{g}_{t-1}^{(j)}|}{\beta_1} = \frac{\beta_2^2|\boldsymbol{g}_{t-2}^{(j)}|}{\beta_1^2} = ... = \frac{\beta_2^k|\boldsymbol{g}_{t-k}^{(j)}|}{\beta_1^k}...$ [2].*

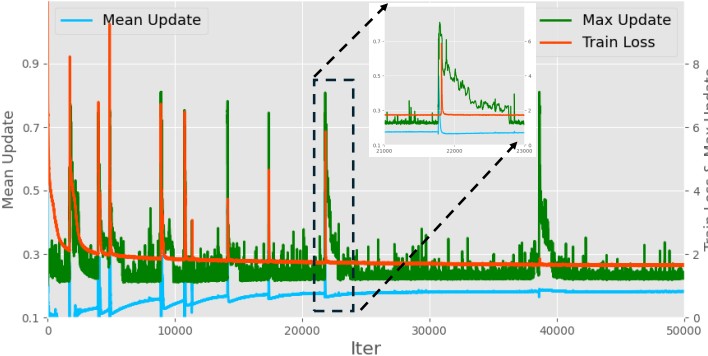

**Figure 2:** Visualization of the mean update (*i.e.*, $\mathrm{Avg}(|\boldsymbol{m}_t^{(j)}|/\sqrt{\boldsymbol{v}_t^{(j)}})$), the maximum update (*i.e.*, $\max_{j \in [d]}(|\boldsymbol{m}_t^{(j)}|/\sqrt{\boldsymbol{v}_t^{(j)}})$), and the training loss over 50,000 iterations during GPT-2 (345M) training on OpenWebText using AdamW ($\beta_1 = 0.9, \beta_2 = 0.999$) with a cosine learning rate schedule. The figure illustrates that all loss spikes are preceded by abrupt increases in the mean update, following a sharp rise in the maximum update. This suggests that a sudden increase in the maximum update for any coordinate can lead to a significant rise in the mean update, which then triggers loss spikes. Moreover, these spikes primarily occur during the early training phase when the learning rate is relatively high. In later stages, as the learning rate decreases, loss spikes become infrequent, even with large maximum updates, like around Iteration 40,000.

Theorem 1 indicates that when Adam is employed, there exists a probability that the update of each element $|\boldsymbol{m}_t^{(j)}|/\sqrt{\boldsymbol{v}_t^{(j)}}$ can reach an excessively large value. For instance, with typical settings of $\beta_1 = 0.9$ and $\beta_2 = 0.999$, the update $|\boldsymbol{m}_t^{(j)}|/\sqrt{\boldsymbol{v}_t^{(j)}}$ could approach its theoretical maximum of $1-\beta_1/\sqrt{1-\beta_2}\sqrt{1-\frac{\beta_1^2}{\beta_2}} \simeq 7.27$, while the normal absolute value of the update is less than 1. While the probability of any specific parameter's update reaching this maximal value is low, the probability that at least one parameter's update reaches this value is high due to the large number of parameters in large models (*e.g.*, LLM). When a parameter's update is excessively large and the learning rate is also high, it is likely to deviate substantially from its intended trajectory. Such deviations may propagate to neighboring parameters through interconnections, triggering a chain reaction that culminates in loss spikes. Supporting examples are clearly illustrated in Figure 1 and Figure 2. This mechanism explains the frequent occurrence of loss spikes during LLM training, particularly in the initial stages when learning rates are higher. Specifically, the probability of loss spikes increases with the size of the LLM. **In conclusion, vanilla Adam poses a significant risk of loss spikes during large-scale**

---

[2] The update $\boldsymbol{m}_t^{(j)}/\sqrt{\boldsymbol{v}_t^{(j)}}$ with respect to $\{\boldsymbol{g}_k^{(j)}\}_{k=1}^t$ is a continuous function. Thus, when most of the signs of $\{\boldsymbol{g}_k^{(j)}\}_{k=1}^t$ are consistent and the secondary condition is nearly satisfied, $\boldsymbol{m}_t^{(j)}/\sqrt{\boldsymbol{v}_t^{(j)}}$ will be close to the theoretical maximum.

model training. Mitigating this problem requires strategies to constrain the maximum update magnitude for each parameter coordinate.

**Remark 1 [Loss spikes during LLM Training].** Encountering loss spikes is a common phenomenon during LLM training (Zeng et al. (2022); Chowdhery et al. (2023); Touvron et al. (2023); Yang et al. (2023)). However, the underlying reasons for this problem were not well explored prior to this. Practitioners had to resort to ad hoc engineering strategies such as skipping some data batches before the spike occurs and restarting training from a nearby checkpoint (Chowdhery et al. (2023); Molybog et al. (2023)), resulting in resource wastage due to frequent rollbacks and checkpointing savings. Some previous works investigated the phenomenon of train instability (Liu et al. (2019)) and loss spikes (Zhu et al. (2023); Zhang & Xu (2023)). (Liu et al. (2019)) demonstrated that the variance of the update $1/\sqrt{\boldsymbol{v}_t^{(j)}}$ is significantly larger, often causing the update to become disproportionately large, but the analysis only works in the early stage. The analyses in (Zhu et al. (2023); Zhang & Xu (2023)) were restricted to either linear models or shallow networks with mean squared error (MSE) loss, using (S)GD as the optimizer. Consequently, it is questionable whether these findings can be directly applied to the context of LLM training. More recently, (Molybog et al. (2023)) suggested that time-domain correlation between gradient estimates of earlier layers contributes to loss spikes during LLM training. The suggested mitigation strategies include lowering the $\epsilon$ value in Adam and reducing the batch size. However, the study itself that these methods are not silver bullets for a fundamental solution. **To the best of our knowledge, the analyses presented above provide the first formal explanation for the frequent occurrence of loss spikes during LLM training.**

## 3 S3 ALGORITHM

Analyzing Adam yields insights guiding the construction of a more effective optimizer for training DNNs: 1) The update of the optimizer need only resemble the sign of the gradient, without strictly adhering to the formulation involving the ratio of first-order gradient momentum to second-order gradient momentum. 2) Minimizing the largest value of the update in the optimizer is crucial to mitigate potential loss spikes.

Recently, several studies (Dozat (2016); Xie et al. (2024); Zhou et al. (2023)) introduced the NAG technique to DNN optimizers, consistently demonstrating faster training convergence and improved inference performance across a broad spectrum of DNNs compared to the standard Adam. Therefore, integrating the NAG technique into optimizers is highly worthwhile.

Given the observations above, we propose a new optimizer, referred to as SoftSignSGD (S3). The detailed implementation of S3 is illustrated in Algorithm 1.

Key characteristics of S3 are summarized below:

*First*, **S3 features a more general sign-like formulation with a flexible $p$-order momentum, extending beyond the fixed $2$-order momentum suggested by the original motivation of Adam.** According to Theorem 2, a large $p$-order momentum enables the use of a larger learning rate, promoting faster convergence and better performance (Kong & Tao (2020)). Moreover, during training, abrupt changes occasionally occur in some coordinates of the gradients, potentially leading to training instability and even divergence without a proper remedy. In such cases, the gradients become more heterogeneous over time. As indicated in Theorem 2, the update $\frac{|\boldsymbol{n}_t|}{\boldsymbol{b}_t(p)}$ of S3 becomes smaller with a large $p$-order momentum, providing a counteractive effect to stabilize the training process. Figure 1 illustrates this behavior. Additionally, the computational cost of optimization becomes nontrivial when training LLMs. Setting $p = 1$ for S3 involves only a computationally light element-wise absolute operation, reducing overall computational overhead.

*Second*, **S3 shares the same exponential moving average coefficient $\beta$ for both $m_t$ and $r_t$, offering the advantages of minimizing the risk of loss spikes and reducing tuning work.** In theory, as demonstrated in Theorem 2, the same $\beta$ guarantees that the largest value of each coordinate of the update $\frac{|\boldsymbol{n}_t|}{\boldsymbol{b}_t(p)}$ is minimized to 1. As discussed in Section 2, this design helps mitigate the loss-spike problem. In practice, this design reduces tuning effort by removing one hyperparameter and lowers computational costs by eliminating the bias correction required by Adam.

*Third*, **S3 introduces the NAG technique to further accelerate training without incurring extra memory costs.** While previous works such as NAdam and Adan have also utilized the NAG technique in their adaptive optimizers, there are significant differences. In S3, NAG is applied to both

---

**Algorithm 1.** SoftSignSGD (S3)

---

1: **Input**: the momentum $\boldsymbol{m}_0 = 0$, $\boldsymbol{s}_0 = 0$, the exponential moving average coefficient $\beta$ within $[0, 1]$, the power factor $p$ within $[1, +\infty)$, and the learning rate sequence $\{\gamma_t\}$.

2: **for** $t = 1, ..., T$ **do**

3:   Randomly sample data and compute the gradient: $\boldsymbol{g}_t \leftarrow \nabla F(\boldsymbol{x}_t; \zeta_t)$

4:   Update the momentum $\boldsymbol{m}_t$: $\boldsymbol{m}_t \leftarrow \beta \boldsymbol{m}_{t-1} + (1-\beta)\boldsymbol{g}_t$

5:   Update the momentum $\boldsymbol{s}_t(p)$: $\boldsymbol{s}_t(p) \leftarrow \beta \boldsymbol{s}_{t-1} + (1-\beta)|\boldsymbol{g}_t|^p$

6:   Compute the Nesterov momentum $\boldsymbol{n}_t$: $\boldsymbol{n}_t \leftarrow \beta \boldsymbol{m}_t + (1-\beta)\boldsymbol{g}_t$

7:   Compute the Nesterov momentum $\boldsymbol{b}_t(p)$: $\boldsymbol{b}_t(p) \leftarrow (\beta \boldsymbol{s}_t(p) + (1-\beta)|\boldsymbol{g}_t|^p)^{1/p}$

8:   Update the model parameter: $\boldsymbol{x}_{t+1} \leftarrow \boldsymbol{x}_t - \gamma_t \frac{\boldsymbol{n}_t}{\boldsymbol{b}_t(p)}$

9: **end for**

---

the numerator and denominator of the update. The Nesterov momentum estimators in S3 follow the NAG (II) formulation, shown to be equivalent to vanilla NAG (I) in Theorem 7. The key advantage of this formulation is that S3 avoids additional memory usage. Conversely, NAdam (Dozat (2016)) only incorporates the Nesterov momentum in the numerator of the update and relies on complex bias-correction operations to stabilize training. Adan (Xie et al. (2024)) also employs Nesterov momentum estimators in both the numerator and the denominator of the update. However, its formulations, akin to NAG (III), demand more memory for storing the previous gradient $\boldsymbol{g}_{t-1}$ and the new momentum $\boldsymbol{r}_k$ compared to Adam. Consequently, *Adan may not be ideal for training LLMs due to its memory demands. Furthermore, it introduces a new momentum coefficient, increasing the need for tuning.*

**Theorem 2.** *The sequences $\{\boldsymbol{n}_t\}$ and $\boldsymbol{b}_t(p)$ are generated S3 in Algorithm 1. If the moving average coefficients for $\boldsymbol{m}_t, \boldsymbol{n}_t$ and $\boldsymbol{s}_t, \boldsymbol{b}_t$ of ar $\beta_1$ and $\beta_2$ which satisfy $\beta_1 < \beta_2^{1/p}$ and $p \geq 1$, it holds that*

*(1). The upper bound of each element of the update $\frac{\boldsymbol{n}_t^{(j)}}{\boldsymbol{b}_t^{(j)}}$ is*

$$\frac{|\boldsymbol{n}_t^{(j)}|}{\boldsymbol{b}_t^{(j)}(p)} \leq \frac{(1-\beta_1)}{(1-\beta_2)^{1/p}\left(1 - \frac{\beta_1^q}{\beta_2^{q/p}}\right)^{1/q}}, \tag{4}$$

*where $\frac{1}{p} + \frac{1}{q} = 1$.*

*(2). When $\beta_1 = \beta_2$, the upper bound of each element of the update $\frac{\boldsymbol{n}_t^{(j)}}{\boldsymbol{b}_t^{(j)}}$ reaches to the smallest 1, i.e., $\frac{|\boldsymbol{n}_t^{(j)}|}{\boldsymbol{b}_t^{(j)}(p)} \leq 1$.*

*(3). Let $1 \leq p_1 \leq p_2$, and then $\boldsymbol{b}_t(p_1) \leq \boldsymbol{b}_t(p_2)$.*

**Theorem 3.** *The three formulations of NAG are listed in the following. Let $\boldsymbol{x}_t = \tilde{\boldsymbol{x}}_t - \gamma\beta\boldsymbol{m}_{t-1}$, the three formulations are equivalent, i.e.,*

$$\text{NAG (I)} : \begin{cases} \boldsymbol{g}_t = \nabla f(\tilde{\boldsymbol{x}}_t - \gamma\beta\boldsymbol{m}_{t-1}; \zeta_t) \\ \boldsymbol{m}_t = \beta\boldsymbol{m}_{t-1} + (1-\beta)\boldsymbol{g}_t \\ \tilde{\boldsymbol{x}}_{t+1} = \tilde{\boldsymbol{x}}_t - \gamma\boldsymbol{m}_t \end{cases}, \tag{5}$$

$$\text{NAG (II)} : \begin{cases} \boldsymbol{g}_t = \nabla f(\boldsymbol{x}_t; \zeta_t) \\ \boldsymbol{m}_t = \beta\boldsymbol{m}_{t-1} + (1-\beta)\boldsymbol{g}_t \\ \boldsymbol{x}_{t+1} = \boldsymbol{x}_t - \gamma(\beta\boldsymbol{m}_t + (1-\beta)\boldsymbol{g}_t) \end{cases}, \tag{6}$$

$$\text{NAG (III)} : \begin{cases} \boldsymbol{g}_t = \nabla f(\boldsymbol{x}_t; \zeta_t) \\ \boldsymbol{m}_t = \beta\boldsymbol{m}_{t-1} + (1-\beta)\boldsymbol{g}_t \\ \boldsymbol{r}_t = \beta\boldsymbol{r}_{t-1} + (1-\beta)(\boldsymbol{g}_t - \boldsymbol{g}_{t-1}) \\ \boldsymbol{x}_{t+1} = \boldsymbol{x}_t - \gamma(\boldsymbol{m}_t + \beta\boldsymbol{r}_t) \end{cases}. \tag{7}$$

*Moreover, if $\tilde{x}_{t+1} \to \tilde{x}_t$ as $m_t \to 0$, $x_t$ will converge to $\tilde{x}_t$.*

## 4 THEORETICAL CONVERGENCE ANALYSIS

To present the theoretical convergence guarantee for S3 (Algorithm 1) to optimize the nonconvex problem in Eq. (1), we first introduce some necessary assumptions.

**Assumption 1.** [Bounded infimum] *There exists a constant $F^*$, the objective function follows $F(\boldsymbol{x}) \geq F^*$ for any $\boldsymbol{x} \in \mathbb{R}^d$.*

**Assumption 2.** [Generalized Smoothness] *There exist constants $L_0, L_1, R \geq 0$, for any $\boldsymbol{x}, \boldsymbol{y} \in \mathbb{R}^d$ with $\|\boldsymbol{x} - \boldsymbol{y}\|_2 \leq R$, the objective function follows,*

$$\|\nabla F(\boldsymbol{y}) - \nabla F(\boldsymbol{x})\|_2 \leq (L_0 + L_1 \|\nabla F(\boldsymbol{x})\|_2) \|\boldsymbol{x} - \boldsymbol{y}\|_2. \tag{8}$$

**Assumption 3.** [Unbias noisy gradient and bounded variance] *There exists a constant $\sigma$. For $\boldsymbol{x}_t \in \mathbb{R}^d$ at any time, the noisy gradient of the objective function obeys follows*

$$\mathbb{E}[\boldsymbol{g}_t] = \mathbb{E}[\nabla f(\boldsymbol{x}_t; \zeta_t)] = \nabla F(\boldsymbol{x}_t), \qquad \mathbb{E}[\|\boldsymbol{g}_t - \nabla F(\boldsymbol{x}_t)\|_2^2] \leq \sigma^2. \tag{9}$$

Under the assumptions above, we then present the theoretical convergence for S3 in Theorem 4.

**Theorem 4.** *$\{x_t\}_{t=1}^T$ is generated by Algorithm 1 under Assumption 1-4. Let the hyperparameters be set as $\beta = 1 - \frac{1}{\sqrt{T}}$ and $\gamma = \frac{1}{L_0 T^{3/4}}$. If $u_t = \frac{|\boldsymbol{n}_t^{(j)}|}{\boldsymbol{b}_t^{(j)}} \geq \frac{1}{U_{\max}}$, then*

$$\frac{1}{T} \sum_{t=1}^T \mathbb{E}[\|\nabla F(\boldsymbol{x}_t)\|_1] \leq \frac{2 L_0 U_{\max}(F(\boldsymbol{x}_1) - F(\boldsymbol{x}^*))}{T^{1/4}} + \frac{4 \beta U_{\max} \sqrt{d} \mathbb{E}[\|\nabla F(\boldsymbol{x}_1)\|_2]}{T^{1/2}}$$
$$+ \frac{4 U_{\max} \sqrt{d} \sigma}{T^{1/4}} + \frac{4 \beta^2 U_{\max} d}{T^{1/4}} + \frac{U_{\max} d}{T^{3/4}}. \tag{10}$$

**Remark 2 [Adopting Weaker Assumption].** The theoretical convergence analysis for S3 in Theorem 4 requires only a general non-uniform smoothness condition (Assumption 2). In contrast, previous works that developed convergence analyses for Adam required stronger assumptions or achieved weaker conclusions. (Chen et al. (2018); Défossez et al. (2020)) proved convergence for non-convex objectives under the assumption that gradients are bounded. (De et al. (2018)) required that the signs of gradients remain the same along the trajectory, despite considering Nesterov acceleration. (Zhang et al. (2022)) assumed uniform $L$-smoothness but only proved convergence to some neighborhood of stationary points with a constant radius. Very recently, (Li et al. (2023); Hong & Lin (2024)) offered convergence bounds for Adam under the general non-uniform smoothness assumption, but the convergence is in probability.

**Remark 3 [Achieving Optimal Convergence Rate].** As illustrated in Theorem 4, the convergence order of S3 is $O(\frac{1}{T^{1/4}})$, consistent with the established lower bound for optimal convergence in non-convex stochastic optimization (Arjevani et al. (2023)).

## 5 EXPERIMENT

We compare S3 with representative optimizers, including SGDM(Robbins & Monro (1951)), AdamW (Loshchilov & Hutter (2017)), NAdam(Dozat (2016)), Adan (Xie et al. (2024)), and Lion (Chen et al. (2023b)), for both vision and language tasks. For vision tasks, we evaluate the classification accuracy by training the CNN-type ResNet-50 (He et al. (2016)) and the Transformer-type ViT-Base (Dosovitskiy et al. (2020)) on ImageNet. In language tasks, we assess the pre-training performance by training GPT-2 (345M) and GPT-2 (7B) (Radford et al. (2019)) on OpenWebText and the refined CommonCrawl. Results on downstream tasks for the pre-trained GPT-2 (345M) and GPT-2 (7B) are provided in the Appendix.

### 5.1 EXPERIMENTS FOR VISION TASKS

Compared with the baseline AdamW, Figure 3(a) and Figure 3 (c) illustrate that S3 exhibits obvious faster convergence and achieves a significantly smaller final training loss. As shown in Figure 3(b), Figure 3(d), and Table 1, S3 achieves test accuracies that are 1.47% and 1.36% higher for training ResNet-50 and ViT-B, respectively. This represents a significant improvement in training speed and

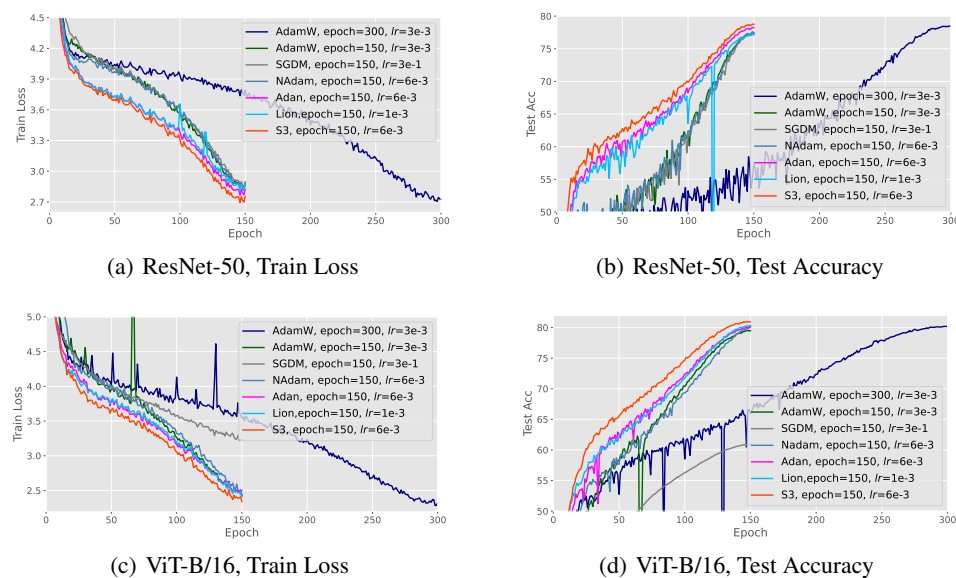

Figure 3: Comparison of train loss and test accuracy on ImageNet for training ReNet-50 and ViT-B/16 with AdamW, SGDM, NAadm, Adan, Lion and S3.

inferencing accuracy. Even when we increase the training epochs by $2\times$ for AdamW, S3 remains comparable and even slightly better. Moreover, AdamW experiences loss spikes during the training of ViT-B, while S3 maintains training stability even with a large learning rate.

In addition, other competitive optimizers, including SGDM, Adan, and Lion, are also evaluated and presented in Figure 3 and Table 1. While SGDM performs comparably to AdamW on the CNN-type ResNet-50, it exhibits poor performance on the Transformer-type ViT-B, consistent with the analyses in Section 2. Due to the introduction of the NAG technique, Adan and Lion outperform AdamW in terms of training speed and test accuracy, yet they still fall short compared to S3. It is noteworthy that Adan and Lion also encounter issues of instability during training.

Table 1: Test accuracy (%) on ImageNet for training ResNet-50 and ViT-B/16 with AdamW, SGDM, Adan, Lion and S3.

| Network | 150 epochs | | | | | | 300 epochs |
| | AdamW | SGDM | NAdam | Adan | Lion | S3 | AdamW |
| --- | --- | --- | --- | --- | --- | --- | --- |
| ResNet-50 | 77.29 | 77.50 | 77.36 | 78.23 | 77.14 | **78.76** | 78.46 |
| ViT-B/16 | 79.52 | 60.99 | 80.31 | 80.11 | 80.32 | **80.93** | 80.13 |

## 5.2 EXPERIMENTS FOR LANGUAGE TASKS

As illustrated in Figure 4 and Table 2, S3 consistently achieves faster train convergence and lower validation perplexity, compared to AdamW. The superiority becomes more obvious as the model size increases. Importantly, the improvement on the 345M model brought by S3 is comparable to that achieved by AdamW with twice the number of steps. This can translate into a significant reduction in the number of steps and total computation needed to reach the same loss level, providing substantial time and cost savings for LLM pre-training. Moreover, while AdamW frequently experiences loss spikes, S3 rarely encounters this issue, even with a learning rate that is $10\times$ larger. Additionally, a large $p$-order momentum for S3 allows a large learning rate, leading to further training acceleration and performance improvement.

Moreover, Lion and Adan are also investigated on the GPT-2 (345M) model. Although Lion converges faster than AdamW at the beginning, it does not showcase superiority in the final validation perplexity. Adan slightly outperforms AdamW in train speed and validation performance, but as analyzed in Section 3, Adan requires more memory and hyperparameters to tune, which is not appealing for pre-training LLMs. Additionally, Adan is also prone to experiencing loss spikes.

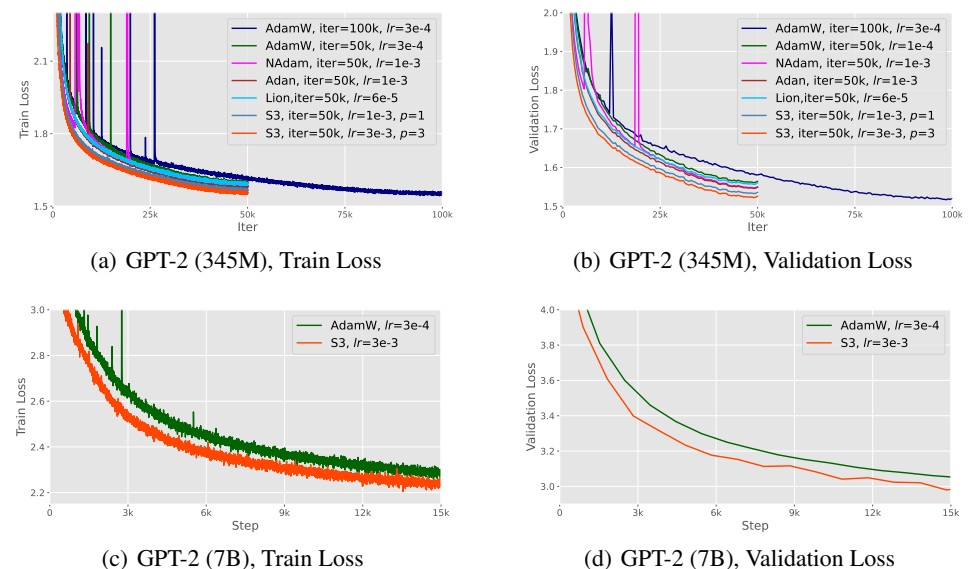

(a) GPT-2 (345M), Train Loss      (b) GPT-2 (345M), Validation Loss

(c) GPT-2 (7B), Train Loss      (d) GPT-2 (7B), Validation Loss

Figure 4: Comparison of train loss and validation loss for pre-training GPT-2 (345M) and GPT-2 (7B) with AdamW, NAdam, Adan, Lion and S3.

Table 2: Validation perplexity (the lower, the better) for training GPT-2 (345M) and GPT-2 (7B).

| Network | Dataset | AdamW ($lr$=3e-4) | NAdam ($lr$=3e-4) | Adan ($lr$=1e-3) | Lion ($lr$=6e-5) | S3 ($p$=3,$lr$=3e-3) | AdamW ($lr$=3e-4) |
|---|---|---|---|---|---|---|---|
| | | 50k steps | | | | | 100k steps |
| GPT-2 (345M) | OpenWebText | 4.78 | 4.71 | 4.69 | 4.76 | **4.59** | 4.57 |
| GPT-2 (7B) | CommonCrawl | 21.13 | - | - | - | **19.69** | - |

## 5.3 ABLATION STUDY

We implement ablation experiments for training ViT-B/16 to clarify the contributions of each modification of S3 over Adam. Figure 5 and Table 3 showcase that employing a large learning rate and sharing the same $\beta$ alone have little or even a negative impact on performance (e.g., Exp. ① vs. Exp. ②, Exp. ① vs. Exp. ③), while their combination results in a notable improvement (e.g., Exp. ②, Exp. ③ vs. Exp. ⑥, Exp. ⑧ vs. Exp. ⑨). In contrast, harnessing a larger $p$ can have an individually beneficial effect on performance (e.g., Exp. ① vs. Exp. ⑤, Exp. ② vs. Exp. ⑧), and the performance gain from the benefits of NAG is more pronounced than other modification (e.g., Exp. ① vs. Exp. ④, Exp. ⑥ vs. Exp. ⑦, Exp. ⑨ vs. Exp. ⑩).

Table 3: Ablation study on test accuracies (%) of S3 for training ViT-B/16.

| Exp. | large $lr$ | same $\beta$ | NAG | flexible $p$ | Test Accuracy |
|---|---|---|---|---|---|
| ① | - | - | - | - | 79.52 (AdamW,$lr$=3e-3) |
| ② | ✔ | - | - | - | 79.45 (AdamW, $lr$=6e-3) |
| ③ | - | ✔ | - | - | 79.48 (S3, $lr$=3e-3, same $\beta$, w/o NAG, $p=2$) |
| ④ | - | - | ✔ | - | 80.17 (S3, $lr$=3e-3, diff. $\beta$, w/ NAG, $p=2$) |
| ⑤ | - | - | - | ✔ | 79.74 (S3, $lr$=3e-3, diff. $\beta$, w/o NAG, $p=3$) |
| ⑥ | ✔ | ✔ | - | - | 80.25 (S3, $lr$=6e-3, same $\beta$, w/o NAG, $p=2$) |
| ⑦ | ✔ | ✔ | ✔ | - | 80.82 (S3, $lr$=6e-3, same $\beta$, w/ NAG, $p=2$) |
| ⑧ | ✔ | - | - | ✔ | 79.98 (S3, $lr$=6e-3, diff. $\beta$, w/o NAG, $p=3$) |
| ⑨ | ✔ | ✔ | - | ✔ | 80.31 (S3, $lr$=6e-3, same $\beta$, w/o NAG, $p=3$) |
| ⑩ | ✔ | ✔ | ✔ | ✔ | 80.93 (S3, $lr$=6e-3, same $\beta$, w/ NAG, $p=3$) |

## 5.4 SENSITIVITY ANALYSIS FOR HYPERPARAMETERS

We perform a grid search to verify the sensitivity to the momentum order $p$ and the momentum coefficient $\beta$ of S3 on ViT-B/16 with 150 training epoches. As shown in Figure 6, all combinations achieve an accuracy of 80.20%+, surpassing the 80.13% accuracy achieved by Adam with 300

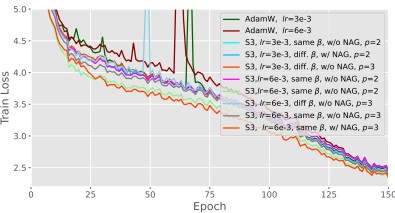
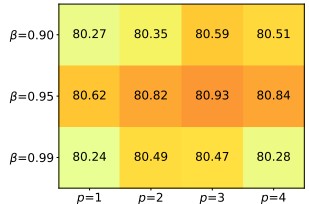

Figure 5: Ablation study on train loss of S3 for training ViT-B/16 on ImageNet.

Figure 6: Impact of the momentum order ($p$) and the momentum coefficient ($\beta$) on the Accuracy of S3 training ViT-B/16 on ImageNet.

training epochs. The performance of S3 is not sensitive when $p > 1$, and $p = 1$ achieves a slightly lower accuracy. However, as pointed out in Section 3, the computation cost becomes lower when $p = 1$. Another interesting phenomenon is that setting $\beta$ to $0.95$ obtains the highest accuracies across different $p$, and $p = 3$ performs well in most cases.

## 5.5 VERIFYING THE REASON FOR LOSS SPIKES FROM ADAM

In this subsection, we further experimentally verify that the claim that the potential overlarge update of Adam with relative large learning rate is underlying reason for loss spikes, as discussed in Section 2. Figure 7 visually illustrates that convergence of vanilla Adam is attained at the baseline learning rate of $3 \times 10^{-4}$ despite sporadic spikes, and more frequent spikes and higher loss are observed at a learning rate of $1 \times 10^{-3}$, with the same iteration count. Moreover, employing a $10\times$ higher learning rate of $3 \times 10^{-3}$ results in premature divergence with pronounced spikes. Noted that all of the phenomenons are aligned with analysis in Section 2. As showcased in Figure 7, naively clipping the Adam update to [-1,1] the range reduces the frequency of loss spikes, but they still occur. This indicates that fine-tuning the clipped value is necessary to balance performance, which complicates the use of clipped updates with tuning a more hyperparameter. In contrast, as we proved in in Theorem 2 in Section 3, when we minimize the maximal update of Adam via $\beta_1 = \beta_2$, loss spikes are completely disappears, which are further verify the correctness of our analyses in Section 2 and Section 3.

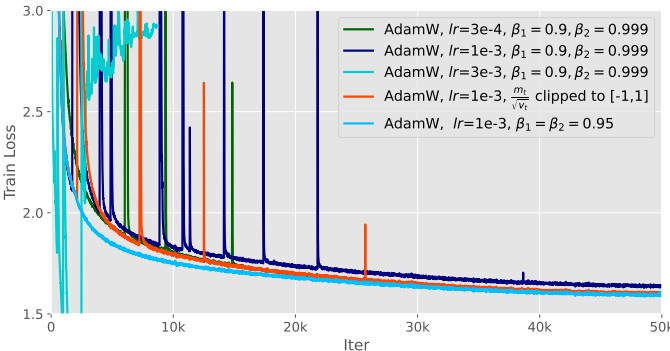

Figure 7: The Loss spikes phenomenons during Training GPT-2 (345M) on OpenWebText using AdamW.

## 6 CONCLUSION AND DISCUSSION

In this paper, we thoroughly examine the strengths and weaknesses of the widely-used optimizer Adam. Building on our analysis, we propose S3, an innovative optimizer that integrates three pivotal improvements over Adam. Comprehensive experiments spanning vision and language tasks showcase S3's accelerated training efficiency and superior inference capabilities. Furthermore, we challenge the conventional belief that Adam's effectiveness stems from simplifying second-order descent, showing instead that its success relies on sign-like descent. This insight paves the way for developing more advanced optimizers. Additionally, We also provide the first theoretical proof of adaptive optimizer convergence from the perspective of sign descent. Most notably, we identify the root cause of loss spikes during LLM training and propose a solution, offering significant benefits for the community in the LLM era.

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

# Appendix

## A  RELATED WORK

**Optimizers in Deep Learning.** Nowadays, Adam has become the dominant optimizer in deep learning. The adaptivity strategy in Adam traces its roots back to earlier optimizers such as Adagrad (Duchi et al. (2011)), RMSprop (Hinton et al. (2012)), and Adadelta (Zeiler (2012)). Beyond Adam, a wide range of variants are proposed (Dozat (2016); Reddi et al. (2018); Loshchilov & Hutter (2017); Zhuang et al. (2020); Shazeer & Stern (2018)). SignSGD, the first sign descent method, was proposed to reduce communication costs in distributed learning (Seide et al. (2014)). Subsequently, (Bernstein et al. (2018); Sun et al. (2023)) provided theoretical convergence for SignSGD and introduced an enhanced version. Chen et al. (2023b) applied an auto ML method to discover the sign descent optimizer Lion. This optimizer demonstrated improved performance with a faster convergence rate on various tasks compared to Adam. Recently, (Liu et al. (2023)) introduced an effective second-order optimizer for LLM pre-training.

**Nesterov's Accelerated Gradient (NAG).** Theoretical demonstrations by (Nesterov (1983; 2013)) indicate that NAG can achieve faster convergence on convex optimization problems compared to vanilla gradient descent, leveraging gradient information at an extrapolation point to anticipate future trends. NAdam by (Dozat (2016)) was the first to incorporate NAG into adaptive optimizers, modifying the first-order momentum of Adam with NAG. Adan by Xie et al. (2024) integrated a equivalence of NAG into both the first and second momentum of Adam, and Win (Zhou et al. (2023)) applied Nesterov acceleration to the update rather than the first and second momentum. Adan and Win outperformed Adam on various tasks, but they require tuning additional hyperparameters and consume more memory, compared to vanilla Adam. Lion (Chen et al. (2023b)), despite being a sign descent method, exhibits a momentum construction similar to NAG (Chen et al. (2023a)). This resemblance could be a contributing factor to its superior speed and performance over Adam.

**Training instability and Loss Spikes in LLM Training.** Training instability and loss spikes are frequently encountered (Zeng et al. (2022); Chowdhery et al. (2023); Touvron et al. (2023); Yang et al. (2023)) during LLM training, posing challenges to further scaling AI systems. To address this issue, practitioners have employed an ad hoc engineering approach, skipping data batches before spikes and restarting training from a nearby checkpoint (Chowdhery et al. (2023)). However, this method requires manual monitoring and intervention, leading to resource wastage. Previous attempts to mitigate instability include embedding norm with BF16, but this comes at a significant performance tax (Scao et al. (2022)). Some researchers found that gradient shrink on the embedding layer reduces loss spikes (Zeng et al. (2022)). Others suggest normalizing the output embedding to lower spike risks (Yang et al. (2023)). (Molybog et al. (2023)) argues that the time-domain correlation between gradient estimates of earlier layers contributes to training loss instability. Mitigation strategies proposed include tuning down the $\epsilon$ value of Adam and reducing batch size. However, it is acknowledged that these methods are not silver bullets for a fundamental solution.

## B  ADDITIONAL EXPERIMENTAL RESULTS

### B.1  TRAINING SETTING

We use the PyTorch vision reference codes [3] to implement vision tasks. For data augmentation, we adhere to the recommended settings in the codes, incorporating RepeatedAugment, AutoAugment Policy (magnitude=9), and Mixup(0.2)/CutMix(1.0) with a probability of 0.5. Additionally, label-smoothing with a value of 0.11 is applied. The batch size is set to 1024 for ResNet-50 and 4096 for ViT-B/16. Regarding the learning rate scheme, we linearly increase it to its peak in the initial 30 epochs and then apply a cosine decay, decreasing it to 0 in the subsequent epochs. Other customized hyperparameters for SGD and AdamW are well-established in the codes, and the settings for Adan and lion are followed the recommendations to train ResNet-50 and ViT-B/16 in their respective official papers (Xie et al. (2024); Chen et al. (2023b)). Since NAdam is similar to AdamW, so all its hyperparameters are also the same as AdamW. Specifically, we list the hyperparameters of all the optimizers as follows:

---

[3]https://github.com/pytorch/vision/tree/main/references/classification

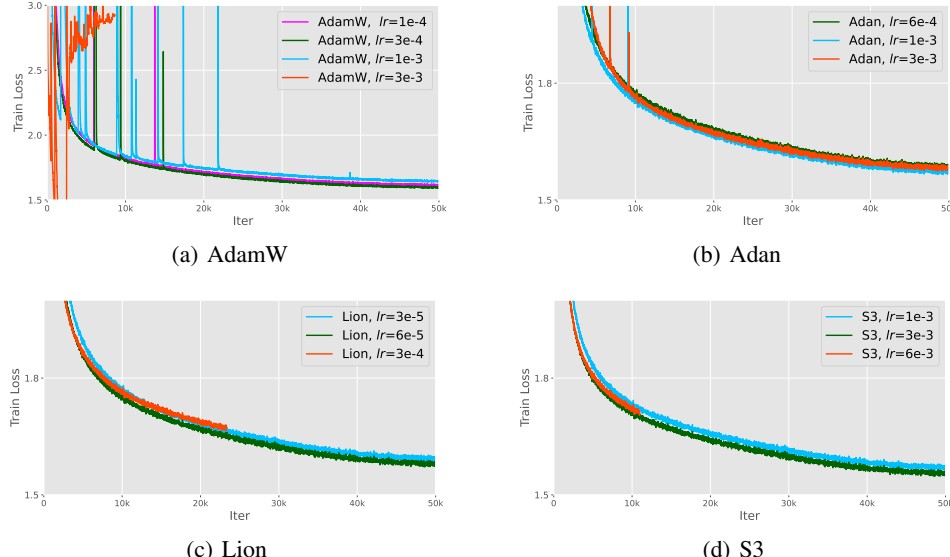

Figure 8: Search the optimal peak learning rates for AdamW, Adan, Lion and S3 pre-training GPT-2 (345M) on OpenWebText.

- For SGD, we use $lr_{\max} = 0.3, \mathrm{mom} = 0.9$ , as is default value in the official codes.
- For AdamW, we utilize $lr_{\max} = 0.003, \beta_1 = 0.9, \beta_2 = 0.999$ to train ResNet-50 and ViT-B/16, as is default value in the official codes and also widely used in other papers (Zhuang et al. (2020); Xie et al. (2024); Chen et al. (2023b)).
- For Adan, we employ $lr_{\max} = 0.015, \beta_1 = 0.98, \beta_2 = 0.92, \beta_3 = 0.99$, per official recommendations (Xie et al. (2024)).
- For Lion, we adopt $lr_{\max} = 0.001, \beta_1 = 0.9, \beta_2 = 0.99$, per official recommendations (iteLion2023).
- For S3, we set $lr_{\max} = 0.006, \beta = 0.95, p = 3$. We conducted a coarse hyperparameters search on ViT-B/16 (Subsection 5.4) and extended the hypermeters to ResNet-50 without further tuning.

We utilize Megatron-LM [4] to implement the language tasks. We use Megatron-LM to implement language tasks. Following the standard GPT-2 protocol, we construct a 345M Transformer decode-only model with the number of layers set to 12, sequence length to 1024, hidden size to 512, and the number of attention heads to 8. To testify the effectiveness of S3 in training a productive LLM model, we additionally construct a large 7B model with the number of layers set to 32, sequence length to 4096, hidden size to 4096, and the number of attention heads to 32. GPT-2 (345M) is trained on OpenWebText with a batch size of 512, and GPT-2 (7B) is trained on the refined CommonCrawl with a batch size of 1024. For S3, we set $p$ to 3 and do not employ the gradient clipping technique. For other optimizers, the gradient clipping threshold is set to 1.0. Regarding the learning rate scheme, we linearly increase it to the peak in the initial $5k$ steps and then decrease to $0.1\times$ of the peak with a cosine decay in the subsequent steps. The peak learning rates of all the optimizers are from coarse search for training GPT-2 (345M)(please refer to Figure 8). Other customized hyperparameters are listed below:

- For AdamW, we utilize $\beta_1 = 0.9, \beta_2 = 0.95$, which are widely used in train LLMs (Zeng et al. (2022); Chowdhery et al. (2023); Touvron et al. (2023); Yang et al. (2023)). Zhuang et al. (2020); Xie et al. (2024); Chen et al. (2023b)).
- For Adan, we employ $\beta_1 = 0.98, \beta_2 = 0.92, \beta_3 = 0.99$, per official recommendations (Xie et al. (2024)) for train LLMs.
- For Lion, we adopt $\beta_1 = 0.95, \beta_2 = 0.98$, per official recommendations (Chen et al. (2023b)) for train LLMs.

---

[4] https://github.com/NVIDIA/Megatron-LM

- For S3, we set $\beta = 0.95, p = 3$. We conducted a coarse hyperparameters search on ViT-B/16 (Subsection 5.4) and extended the hypermeters to train LLMs without further tuning.

Notably, we followed the weight decay adjustment strategy outlined in the paper (Chen et al. (2023b)). Specifically, we used the product of the peak learning rate ($lr_{\mathrm{Adam}}$) and the weight decay ($\lambda_{\mathrm{Adam}}$) from AdamW as a constant. For other optimizers, we just determine the peak leaning rate, and the weight decay was derived directly using the formula $\lambda = \frac{lr_{\mathrm{Adam}}\lambda_{\mathrm{Adam}}}{lr}$. Importantly, the the baseline peak learning rates and weight decays of Adam for training our ResNet-50 and ViT-B-16 are also the same with those reported in (Chen et al. (2023b)), while that for GPT-2 are the same with the paper on Llama (Touvron et al. (2023)).

Table 4: Ablation study on validation perplexity of S3 for training GPT-2(345M).

| Exp. | large $lr$ | same $\beta$ | NAG | flexible $p$ | Validation perplexity |
|:---:|:---:|:---:|:---:|:---:|:---|
| ① | - | - | - | - | 4.78 (AdamW,$lr$=3e-4) |
| ② | ✔ | - | - | - | 4.97 (AdamW, $lr$=1e-3) |
| ③ | - | ✔ | - | - | 4.77 (S3, $lr$=3e-4, same $\beta$, w/o NAG, $p = 2$) |
| ④ | ✔ | ✔ | - | - | 4.67 (S3, $lr$=1e-3, same $\beta$, w/o NAG, $p = 2$) |
| ⑤ | ✔ | ✔ | ✔ | - | 4.64 (S3, $lr$=1e-3, same $\beta$, w/ NAG, $p = 2$) |
| ⑥ | ✔ | ✔ | ✔ | ✔ | 4.60 (S3, $lr$=3e-3, same $\beta$, w/ NAG, $p = 3$) |

## B.2 ADDITIONAL ABLATION STUDY

We also implement ablation experiments for training GPT-2(345M). Figure 10 and Table 4 showcase that employing a large learning rate and sharing the same $\beta$ alone have little or even a negative impact on performance (e.g., Exp. ① vs. Exp. ②, Exp. ① vs. Exp. ③), while their combination results in a notable improvement (e.g., Exp. ②, Exp. ③ vs. Exp. ④). In contrast, the performance gain from the benefits of NAG is more pronounced (e.g., Exp. ④ vs. Exp. ⑤), and harnessing a larger $p$ can also have an individually beneficial effect on performance (e.g., Exp. ⑤ vs. Exp. ⑥),

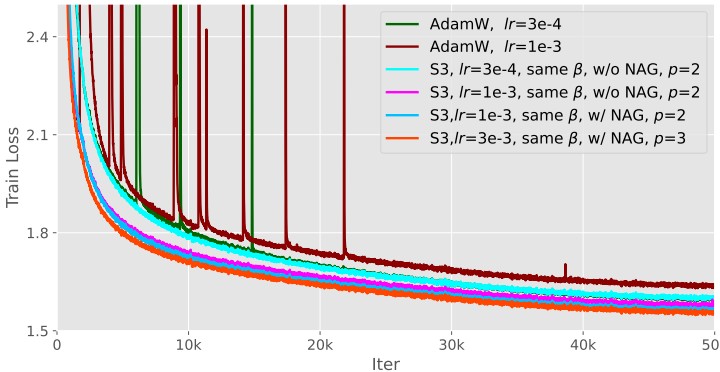

Figure 9: Ablation study on train loss of S3 for training GPT-2(345M) on OpenWebText.

## B.3 DOWNSTREAM EVALUATION FOR LANGUAGE TASKS

To further validate the effectiveness of the proposed optimizers, we conducted evaluation experiments on pre-trained GPT-2 models, specifically GPT-2 (345M) and GPT-2 (7B), using downstream reasoning benchmarks from OpenCompass [5]. As depicted in Figure 10, the results demonstrate that S3 consistently outperforms Adam across the majority of benchmarks. This superiority is evident in the improved downstream accuracy, indicating that the lower validation loss achieved by S3 translates into enhanced performance on these reasoning tasks.

An interesting observation is that the superiority of S3 becomes more pronounced as the model size becomes large. This suggests that the benefits of S3 extend beyond its effectiveness with smaller models, showcasing its scalability and adaptability to larger and more complex architectures.

---

[5]https://github.com/open-compass/opencompass

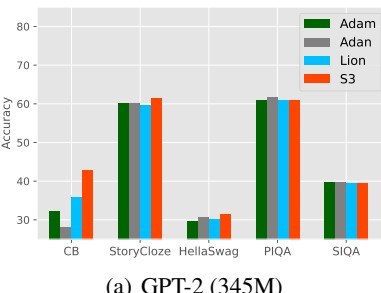
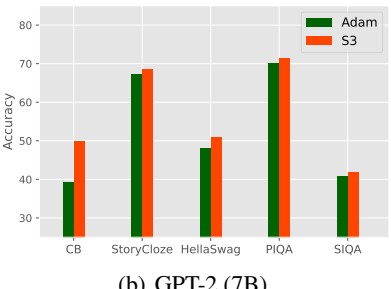

(a) GPT-2 (345M)  (b) GPT-2 (7B)

Figure 10:  Zero-shot evaluation of the pre-trained GPT-2 (345M) and GPT-2 (7B) with AdamW, Adan, aLion, and S3 on downstream reasoning tasks.

It is essential to acknowledge that the GPT-2 (345M) model, along with its training dataset, is relatively small. Consequently, the pre-trained models may lack the inherent capabilities needed to perform well on downstream benchmarks, regardless of the optimizer used. Consequently, the accuracies achieved by GPT-2 (345M) with these optimizers may exhibit a degree of randomness due to the model's inherent limitations in handling more complex tasks with a smaller scale.

### B.4    Visualization of Gradient Norms

As shown in Figure 11, the gradient norms can differ by several orders of magnitude across different layers, and within the same layer, the gradient norms can vary by more than 30 times. This significant variation highlights the challenge of maintaining consistent update magnitudes during the training process.

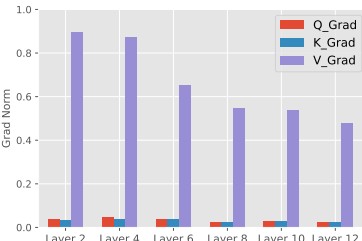

Figure 11:  Visualization of gradient norms within different layers in ViT-B/16 at initialization.

## C    Theoretical Proofs

### C.1    Proof of Theorem 1

**Proof.** Recalling Eq. (2), we know

$$
\boldsymbol{m}_t^{(j)} = \frac{1 - \beta_1}{1 - \beta_1^t} \sum_{k=1}^{t} \beta_1^{t-k} \boldsymbol{g}_k^{(j)}
$$

$$
\boldsymbol{v}_t^{(j)} = \frac{1 - \beta_2}{1 - \beta_2^t} \sum_{k=1}^{t} \beta_2^{t-k} (\boldsymbol{g}_k^{(j)})^2.
$$

(11)

Then,

$$
\begin{aligned}
\frac{|\boldsymbol{m}_t^{(j)}|}{\sqrt{\boldsymbol{v}_t^{(j)}}} &= \frac{(1-\beta_1)\sqrt{1-\beta_2^t}}{(1-\beta_1^t)\sqrt{1-\beta_2}} \cdot \frac{|\sum_{k=1}^{t} \beta_1^{t-k} \boldsymbol{g}_k^{(j)}|}{\sqrt{\sum_{k=1}^{t} \beta_2^{t-k} (\boldsymbol{g}_k^{(j)})^2}} \\
&\overset{(i)}{\leq} \frac{(1-\beta_1)\sqrt{1-\beta_2^t}}{(1-\beta_1^t)\sqrt{1-\beta_2}} \cdot \frac{\sum_{k=1}^{t} \beta_1^{t-k} |\boldsymbol{g}_k^{(j)}|}{\sqrt{\sum_{k=1}^{t} \beta_2^{t-k} (\boldsymbol{g}_k^{(j)})^2}} \\
&\overset{(ii)}{\leq} \frac{(1-\beta_1)\sqrt{1-\beta_2^t}}{(1-\beta_1^t)\sqrt{1-\beta_2}} \cdot \frac{\sqrt{\sum_{k=1}^{t} \beta_2^{t-k} (\boldsymbol{g}_k^{(j)})^2} \sqrt{\sum_{k=1}^{t} \frac{\beta_1^{2(t-k)}}{\beta_2^{t-k}}}}{\sqrt{\sum_{k=1}^{t} \beta_2^{t-k} (\boldsymbol{g}_k^{(j)})^2}} \\
&= \frac{(1-\beta_1)\sqrt{1-\beta_2^t}}{(1-\beta_1^t)\sqrt{1-\beta_2}} \cdot \sqrt{\sum_{k=1}^{t} \frac{\beta_1^{2(t-k)}}{\beta_2^{t-k}}} \\
&\overset{(iii)}{=} \frac{(1-\beta_1)\sqrt{1-\beta_2^t}\sqrt{1-(\frac{\beta_1^2}{\beta_2})^t}}{(1-\beta_1^t)\sqrt{1-\beta_2}\sqrt{1-\frac{\beta_1^2}{\beta_2}}} \\
&\simeq \frac{1-\beta_1}{\sqrt{1-\beta_2}\sqrt{1-\frac{\beta_1^2}{\beta_2}}},
\end{aligned}
\tag{12}
$$

where $(i)$ holds due to the fact $|\boldsymbol{a}^{(j)} + \boldsymbol{b}^{(j)}| \leq |\boldsymbol{a}^{(j)}| + |\boldsymbol{b}^{(j)}|$; $(ii)$ holds due to Cauchy-Schiwaz inequality; $(iii)$ holds due to $\beta_1^2 \leq \beta_2$. $\frac{|\boldsymbol{m}_t^{(j)}|}{\sqrt{\boldsymbol{v}_t^{(j)}}}$ reach to the largest value if the signs of $\{\boldsymbol{g}_t^{(j)}, \boldsymbol{g}_{t-1}^{(j)}, ...\}$ are the same and $|\boldsymbol{g}_t^{(j)}| = \frac{\beta_2 |\boldsymbol{g}_{t-1}^{(j)}|}{\beta_1} = \frac{\beta_2^2 |\boldsymbol{g}_{t-2}^{(j)}|}{\beta_1^2} = ... = \frac{\beta_2^k |\boldsymbol{g}_{t-k}^{(j)}|}{\beta_1^k} ....$

### C.2  Proof of Theorem 2

**Proof.** (1). According to S3 in Algorithm 1, we have

$$
\begin{aligned}
\boldsymbol{n}_t^{(j)} &= (1-\beta_1)\left(\sum_{k=1}^{t} \beta_1^{t-k+1} \boldsymbol{g}_k^{(j)} + \boldsymbol{g}_t^{(j)}\right) \\
\boldsymbol{b}_t^{(j)}(p) &= \left((1-\beta_2)\left(\sum_{k=1}^{t} \beta_2^{t-k+1} |\boldsymbol{g}_k^{(j)}|^p + |\boldsymbol{g}_t^{(j)}|^p\right)\right)^{1/p}
\end{aligned}
\tag{13}
$$

Then, assuming $q$ satisfies $\frac{1}{p} + \frac{1}{q} = 1$, we obtain

$$
\begin{aligned}
\frac{|\boldsymbol{n}_t^{(j)}|}{\boldsymbol{b}_t^{(j)}(p)} &= \frac{1-\beta_1}{(1-\beta_2)^{1/p}} \cdot \frac{|\sum_{k=1}^t \beta_1^{t-k+1} \boldsymbol{g}_k^{(j)} + \boldsymbol{g}_t^{(j)}|}{\left(\sum_{k=1}^t \beta_2^{t-k+1}|\boldsymbol{g}_k^{(j)}|^p + |\boldsymbol{g}_t^{(j)}|^p\right)^{1/p}} \\[2mm]
&\overset{(i)}{\leq} \frac{1-\beta_1}{(1-\beta_2)^{1/p}} \cdot \frac{\sum_{k=1}^t \beta_1^{t-k+1}|\boldsymbol{g}_k^{(j)}| + |\boldsymbol{g}_t^{(j)}|}{\left(\sum_{k=1}^t \beta_2^{t-k+1}|\boldsymbol{g}_k^{(j)}|^p + |\boldsymbol{g}_t^{(j)}|^p\right)^{1/p}} \\[2mm]
&\overset{(ii)}{\leq} \frac{1-\beta_1}{(1-\beta_2)^{1/p}} \cdot \frac{\left(\sum_{k=1}^t \beta_2^{t-k+1}|\boldsymbol{g}_k^{(j)}|^p + |\boldsymbol{g}_t^{(j)}|^p\right)^{1/p} \left(\sum_{k=1}^t \left(\frac{\beta_1^{(t-k+1)}}{\beta_2^{\frac{1}{p}(t-k+1)}}\right)^q + 1\right)^{1/q}}{\left(\sum_{k=1}^t \beta_2^{t-k+1}|\boldsymbol{g}_k^{(j)}|^p + |\boldsymbol{g}_t^{(j)}|^p\right)^{1/p}} \\[2mm]
&= \frac{1-\beta_1}{(1-\beta_2)^{1/p}} \cdot \left(\sum_{k=1}^t \left(\frac{\beta_1^{(t-k+1)}}{\beta_2^{\frac{1}{p}(t-k+1)}}\right)^q + 1\right)^{1/q} \\[2mm]
&\overset{(iii)}{\leq} \frac{1-\beta_1}{(1-\beta_2)^{1/p}\left(1 - \frac{\beta_1^q}{\beta_2^{q/p}}\right)^{1/q}},
\end{aligned}
\tag{14}
$$

where $(i)$ holds due to the fact $|a+b| \leq |a| + |b|$; $(ii)$ holds due to Hölder inequality $\sum_{i=1}^s a_i b_i \leq (\sum_{i=1}^s a_i^p)^{1/p}(\sum_{i=1}^s b_i^q)^{1/q}$ if $\frac{1}{p} + \frac{1}{q} = 1$ and $p, q \geq 1$; $(iii)$ holds due to $\beta_1 \leq \beta_2^{1/p}$.

(2). The upper bound of each element of the update $\frac{\boldsymbol{n}_t^{(j)}}{\boldsymbol{b}_t^{(j)}}$ can be

$$
\begin{aligned}
\frac{1-\beta_1}{(1-\beta_2)^{1/p}\left(1 - \frac{\beta_1^q}{\beta_2^{q/p}}\right)^{1/q}} &\overset{(i)}{\geq} \frac{1-\beta_1}{\frac{1}{p}(1-\beta_2) + \frac{1}{q}\left(1 - \frac{\beta_1^q}{\beta_2^{q/p}}\right)} \\[2mm]
&\overset{(ii)}{=} \frac{1-\beta_1}{1 - \left(\frac{\beta_2}{p} + \frac{\beta_1^q}{q\beta_2^{q/p}}\right)} \\[2mm]
&\overset{(iii)}{\geq} \frac{1-\beta_1}{1 - \beta_2^{1/p}\frac{\beta_1}{\beta_2^{1/p}}} \\[2mm]
&= \frac{1-\beta_1}{1-\beta_1} \\[2mm]
&= 1,
\end{aligned}
\tag{15}
$$

where $(i)$ holds due to Young's inequality $\frac{a}{p} + \frac{b}{q} \geq a^{1/p}b^{1/q}$; $(ii)$ holds owing to $\frac{1}{p} + \frac{1}{q} = 1$; $(iii)$ holds resulting from Young's inequality again.

Notably, according to the property of Young's inequality, the equality in $(i)$ and $(iii)$ can be reached, if and only if

$$
\beta_2 = \frac{\beta_1^q}{\beta_2^{q/p}} \quad \Rightarrow \quad \beta_1^q = \beta_2^{1+\frac{q}{p}} \quad \Rightarrow \quad \beta_1^q = \beta_2^{1+(1-\frac{1}{q})q} \quad \Rightarrow \quad \beta_1^q = \beta_2^q \quad \Rightarrow \quad \beta_1 = \beta_2 \tag{16}
$$

Therefore, when $\beta_1 = \beta_2$, the upper bound of each element of the update $\frac{\boldsymbol{n}_t^{(j)}}{\boldsymbol{b}_t^{(j)}}$ reaches to the smallest 1.

(3). Following S3 in Algorithm 1, we have

$$\boldsymbol{b}_t^{(j)}(p_1) = \left((1-\beta)\left(\sum_{k=1}^{t}\beta^{t-k+1}|\boldsymbol{g}_k^{(j)}|^{p_1} + |\boldsymbol{g}_t^{(j)}|^{p_1}\right)\right)^{1/p_1}$$

$$\boldsymbol{b}_t^{(j)}(p_2) = \left((1-\beta)\left(\sum_{k=1}^{t}\beta^{t-k+1}|\boldsymbol{g}_k^{(j)}|^{p_2} + |\boldsymbol{g}_t^{(j)}|^{p_2}\right)\right)^{1/p_2}. \tag{17}$$

Denoting $r = \frac{p_2}{p_1}$, we then obtain

$$\begin{aligned}
(\boldsymbol{b}_t^{(j)}(p_1))^{p_2} = (\boldsymbol{b}_t^{(j)}(p_1))^{rp_1} &= \left((1-\beta)\left(\sum_{k=1}^{t}\beta^{t-k+1}|\boldsymbol{g}_k^{(j)}|^{p_1} + |\boldsymbol{g}_t^{(j)}|^{p_1}\right)\right)^{r} \\
&\leq (1-\beta)\left(\sum_{k=1}^{t}\beta^{t-k+1}|\boldsymbol{g}_k^{(j)}|^{rp_1} + |\boldsymbol{g}_t^{(j)}|^{rp_1}\right) \\
&= (1-\beta)\left(\sum_{k=1}^{t}\beta^{t-k+1}|\boldsymbol{g}_k^{(j)}|^{p_2} + |\boldsymbol{g}_t^{(j)}|^{p_2}\right) \\
&= (\boldsymbol{b}_t^{(j)}(p_2))^{p_2},
\end{aligned} \tag{18}$$

where the inequality holds due to Jensen's inequality and the fact $(1-\beta)(\sum_{k=1}^{t}\beta^{t-k+1} + 1) < 1$.

## C.3 Proof of Theorem 3

**Proof.** We first deduce from ASGD(I) to ASGD(II). According to (I), we have

$$\begin{aligned}
\tilde{\boldsymbol{x}}_{t+1} &= \tilde{\boldsymbol{x}}_t - \gamma\boldsymbol{m}_t \\
&= \tilde{\boldsymbol{x}}_t - \gamma(\beta\boldsymbol{m}_{t-1} + (1-\beta)\nabla f(\tilde{\boldsymbol{x}}_t - \gamma\beta\boldsymbol{m}_{t-1}; \zeta_t))
\end{aligned} \tag{19}$$

Subtracting $\gamma\beta\boldsymbol{m}_t$ on both sides, we obtain

$$\tilde{\boldsymbol{x}}_{t+1} - \gamma\beta\boldsymbol{m}_t = \tilde{\boldsymbol{x}}_t - \gamma\beta\boldsymbol{m}_{t-1} - \gamma(\beta\boldsymbol{m}_t - (1-\beta)\nabla f(\tilde{\boldsymbol{x}}_t - \gamma\beta\boldsymbol{m}_{t-1}; \zeta_t)) \tag{20}$$

Setting $\boldsymbol{x}_t = \tilde{\boldsymbol{x}}_t - \gamma\beta\boldsymbol{m}_{t-1}$, we further have

$$\boldsymbol{x}_{t+1} = \boldsymbol{x}_t - \gamma(\beta\boldsymbol{m}_t + (1-\beta)\nabla f(\boldsymbol{x}_t; \zeta_t)) \tag{21}$$

Thus, ASGD(I) becomes ASGD(II).

Then, we deduce from ASGD(III). Denoting $\boldsymbol{n}_t = \beta\boldsymbol{m}_t + (1-\beta)\boldsymbol{g}_t$, we have

$$\begin{aligned}
\boldsymbol{n}_t - \beta\boldsymbol{n}_{t-1} &= \beta\boldsymbol{m}_t + (1-\beta)\boldsymbol{g}_t - \beta\boldsymbol{n}_{t-1} \\
&= (1-\beta)\boldsymbol{g}_t + \beta(\beta\boldsymbol{m}_{t-1} + (1-\beta)\boldsymbol{g}_t) - \beta\boldsymbol{n}_{t-1} \\
&= (1-\beta)\boldsymbol{g}_t + \beta(\beta\boldsymbol{m}_{t-1} + (1-\beta)\boldsymbol{g}_t) - \beta(\beta\boldsymbol{m}_{t-t} + (1-\beta)\boldsymbol{g}_{t-1}) \\
&= (1-\beta)\boldsymbol{g}_t + \beta(1-\beta)(\boldsymbol{g}_t - \boldsymbol{g}_{t-1}) \\
&= (1-\beta)(\boldsymbol{g}_t + \beta(\boldsymbol{g}_t - \boldsymbol{g}_{t-1})).
\end{aligned} \tag{22}$$

It indicates $\boldsymbol{n}_t = \boldsymbol{m}_t + \beta\boldsymbol{r}_t$ where $\boldsymbol{m}_t = \beta\boldsymbol{m}_{t-1} + (1-\beta)\boldsymbol{g}_t$ and $\boldsymbol{r}_t = \beta\boldsymbol{r}_{t-1} + (1-\beta)(\boldsymbol{g}_t - \boldsymbol{g}_{t-1})$. Therefore, ASGD (II) is equivalent to ASGD (III).

## C.4 Auxiliary Lemma

**Lemma 5.** *Under Assumption 2, for any* $\boldsymbol{x}, \boldsymbol{y} \in \mathbb{R}^d$ *with* $\|\boldsymbol{x} - \boldsymbol{y}\|_2 \leq R$, *the function obeys*

$$F(\boldsymbol{y}) \leq F(\boldsymbol{x}) + \langle\nabla F(\boldsymbol{x}), \boldsymbol{y} - \boldsymbol{x}\rangle + \frac{L_0 + L_1\|\nabla F(\boldsymbol{x})\|_2}{2}\|\boldsymbol{y} - \boldsymbol{x}\|_2^2. \tag{23}$$

**Proof.** For any $\boldsymbol{x}, \boldsymbol{y} \in \mathbb{R}^d$ with $\|\boldsymbol{x} - \boldsymbol{y}\|_2 \leq R$, we have

$$
\begin{aligned}
F(\boldsymbol{y}) =& F(\boldsymbol{x}) + \int_0^1 \langle \nabla F(\boldsymbol{x} + t(\boldsymbol{y} - \boldsymbol{x})), \boldsymbol{y} - \boldsymbol{x} \rangle dt \\
=& F(\boldsymbol{x}) + \langle \nabla F(\boldsymbol{x}), \boldsymbol{y} - \boldsymbol{x} \rangle + \int_0^1 \langle \nabla F(\boldsymbol{x} + t(\boldsymbol{y} - \boldsymbol{x})) - \nabla F(\boldsymbol{x}), \boldsymbol{y} - \boldsymbol{x} \rangle dt \\
\overset{(i)}{\leq}& F(\boldsymbol{x}) + \langle \nabla F(\boldsymbol{x}), \boldsymbol{y} - \boldsymbol{x} \rangle + \int_0^1 \|\nabla F(\boldsymbol{x} + t(\boldsymbol{y} - \boldsymbol{x})) - \nabla F(\boldsymbol{x})\|_2 \|\boldsymbol{y} - \boldsymbol{x}\|_2 dt \quad (24) \\
\overset{(ii)}{\leq}& F(\boldsymbol{x}) + \langle \nabla F(\boldsymbol{x}), \boldsymbol{y} - \boldsymbol{x} \rangle + (L_0 + L_1 \nabla F(\boldsymbol{x})) \|\boldsymbol{y} - \boldsymbol{x}\|_2^2 \int_0^1 t d_t \\
=& F(\boldsymbol{x}) + \langle \nabla F(\boldsymbol{x}), \boldsymbol{y} - \boldsymbol{x} \rangle + \frac{L_0 + L_1 \|\nabla F(\boldsymbol{x})\|}{2} \|\boldsymbol{y} - \boldsymbol{x}\|_2^2,
\end{aligned}
$$

where $(i)$ holds due to Cauchy-Schwarz inequality, and $(ii)$ holds due to Assumption 2.

## C.5  PROOF OF THEOREM 4

**Proof.** Following Lemma 5 with $\boldsymbol{x}_{t+1} \to \boldsymbol{y}$ and $\boldsymbol{x}_t \to \boldsymbol{x}$, we have

$$
F(\boldsymbol{x}_{t+1}) \leq F(\boldsymbol{x}_t) + \langle \nabla F(\boldsymbol{x}_t), \boldsymbol{x}_{t+1} - \boldsymbol{x}_t \rangle + \frac{L_0 + L_1 \|\nabla F(\boldsymbol{x}_t)\|_2}{2} \|\boldsymbol{x}_{t+1} - \boldsymbol{x}_t\|_2^2. \quad (25)
$$

Recalling the update rule $\boldsymbol{x}_{t+1} = \boldsymbol{x}_t - \gamma \frac{\boldsymbol{n}_t}{\boldsymbol{b}_t} = \boldsymbol{x}_t - \gamma \frac{|\boldsymbol{n}_t|}{\boldsymbol{b}_t} \circ \frac{\boldsymbol{n}_t}{|\boldsymbol{n}_t|} = \boldsymbol{x}_t - \gamma \boldsymbol{u}_t \circ \mathrm{Sign}(\boldsymbol{n}_t)$, we further obtain

$$
\begin{aligned}
F(\boldsymbol{x}_{t+1}) \leq& F(\boldsymbol{x}_t) - \langle \nabla F(\boldsymbol{x}_t), \gamma \boldsymbol{u}_t \circ \mathrm{Sign}(\boldsymbol{n}_t) \rangle + \frac{(L_0 + L_1 \|\nabla F(\boldsymbol{x}_t)\|_2)\gamma^2}{2} \|\boldsymbol{u}_t\|_2^2 \\
=& F(\boldsymbol{x}_t) - \langle \nabla F(\boldsymbol{x}_t), \gamma \boldsymbol{u}_t \circ \mathrm{Sign}(\nabla F(\boldsymbol{x}_t)) \rangle + \underbrace{\langle \nabla F(\boldsymbol{x}_t), \gamma \boldsymbol{u}_t \circ (\mathrm{Sign}(\nabla F(\boldsymbol{x}_t)) - \mathrm{Sign}(\boldsymbol{n}_t)) \rangle}_{T_1} \\
& + \frac{(L_0 + L_1 \|\nabla F(\boldsymbol{x}_t)\|)\gamma^2}{2} \|\boldsymbol{u}_t\|_2^2.
\end{aligned}
$$
$$(26)$$

There are two cases for each element of $T_1$. If $\mathrm{Sign}(\nabla F(\boldsymbol{x}_t)^{(j)}) = \mathrm{Sign}(\boldsymbol{n}_t^{(j)})$, $\nabla F(\boldsymbol{x}_t)^{(j)} \cdot \boldsymbol{u}_t^{(j)} \cdot \left( \mathrm{Sign}(\nabla F(\boldsymbol{x}_t))^{(j)} - \mathrm{Sign}(\boldsymbol{n}_t^{(j)}) \right) = 0$. If $\mathrm{Sign}(\nabla F(\boldsymbol{x}_t)^{(j)}) \neq \mathrm{Sign}(\boldsymbol{n}_t^{(j)})$, $\nabla F(\boldsymbol{x}_t)^{(j)} \cdot \boldsymbol{u}_t^{(j)} \cdot \left( \mathrm{Sign}(\nabla F(\boldsymbol{x}_t))^{(j)} - \mathrm{Sign}(\boldsymbol{n}_t^{(j)}) \right) = 2\boldsymbol{u}_t^{(j)} |\nabla F(\boldsymbol{x}_t)^{(j)}| \leq 2\boldsymbol{u}_t^{(j)} |\nabla F(\boldsymbol{x}_t)^{(j)} - \boldsymbol{n}_t^{(j)}|$, hence $T_1 = 2 \sum_{j=1}^d \boldsymbol{u}_t^{(j)} |\nabla F(\boldsymbol{x}_t)^{(j)} - \boldsymbol{n}_t^{(j)}|$.

Rearranging Eq. (26), we have

$$
\begin{aligned}
F(\boldsymbol{x}_{t+1}) \leq& F(\boldsymbol{x}_t) - \langle \nabla F(\boldsymbol{x}_t), \gamma \boldsymbol{u}_t \circ \mathrm{Sign}(\nabla F(\boldsymbol{x}_t)) \rangle + 2\gamma \sum_{j=1}^d \boldsymbol{u}_t^{(j)} |\nabla F(\boldsymbol{x}_t)^{(j)} - \boldsymbol{n}_t^{(j)}| \\
& + \frac{(L_0 + L_1 \|\nabla F(\boldsymbol{x}_t)\|_2)\gamma^2}{2} \|\boldsymbol{u}_t\|_2^2 \\
\leq& F(\boldsymbol{x}_t) - \gamma u_{\min} \|\nabla F(\boldsymbol{x}_t)\|_1 + 2\gamma \|\boldsymbol{n}_t - \nabla F(\boldsymbol{x}_t)\|_1 + \frac{\gamma^2 d(L_0 + L_1 \|\nabla F(\boldsymbol{x}_t)\|_2)}{2} \\
\leq& F(\boldsymbol{x}_t) - \gamma u_{\min} \|\nabla F(\boldsymbol{x}_t)\|_1 + 2\gamma \sqrt{d} \|\boldsymbol{n}_t - \nabla F(\boldsymbol{x}_t)\|_2 + \frac{\gamma^2 d(L_0 + L_1 \|\nabla F(\boldsymbol{x}_t)\|_1)}{2}
\end{aligned}
$$
$$(27)$$

where the second inequality holds due to $0 < u_{\min} \leq \boldsymbol{u}_t^{(j)} \leq 1$, and the third inequality holds owing to the fact $\|\boldsymbol{a}\|_2 \leq \|\boldsymbol{a}\|_1 \leq \sqrt{d}\|\boldsymbol{a}\|_2$ for any $\boldsymbol{a} \in \mathbb{R}^d$.

Summing over 1 to $T$ and taking expectation on it, we have

$$\frac{(u_{\min} - \frac{\gamma d L_1}{2})}{T} \sum_{t=1}^{T} \mathbb{E}[\|\nabla F(\boldsymbol{x}_t)\|_1] \leq \frac{F(\boldsymbol{x}_1) - F(\boldsymbol{x}^*)}{\gamma T} + \frac{2\sqrt{d}}{T} \sum_{t=1}^{T} \mathbb{E}[\|\boldsymbol{n}_t - \nabla F(\boldsymbol{x}_t)\|_2] + \frac{\gamma d L_0}{2} \tag{28}$$

Recalling $\boldsymbol{m}_t = \beta \boldsymbol{m}_{t-1} + (1-\beta)\boldsymbol{g}_t$, we obtain

$$\begin{aligned}
\boldsymbol{m}_t - \nabla F(\boldsymbol{x}_t) &= (\beta \boldsymbol{m}_{t-1} + (1-\beta)\boldsymbol{g}_t) - \nabla F(\boldsymbol{x}_t) \\
&= \beta(\boldsymbol{m}_{t-1} - \nabla F(\boldsymbol{x}_{t-1})) + (1-\beta)(\boldsymbol{g}_t - \nabla F(\boldsymbol{x}_t)) - \beta(\nabla F(\boldsymbol{x}_t) - \nabla F(\boldsymbol{x}_{t-1})).
\end{aligned} \tag{29}$$

Utilizing recursion, we further have

$$\boldsymbol{m}_t - \nabla F(\boldsymbol{x}_t) = -\beta^t \nabla F(\boldsymbol{x}_1) + (1-\beta) \sum_{k=1}^{t} \beta^{t-k}(\boldsymbol{g}_k - \nabla F(\boldsymbol{x}_k)) - \sum_{k=1}^{t} \beta^{t-k+1}(\nabla F(\boldsymbol{x}_k) - \nabla F(\boldsymbol{x}_{k-1})), \tag{30}$$

where $\boldsymbol{m}_1 - \nabla F(\boldsymbol{x}_1) = -\beta_1 \nabla F(\boldsymbol{x}_1) + (1-\beta_1)(\boldsymbol{g}_1 - \nabla F(\boldsymbol{x}_1))$ due to $\boldsymbol{m}_0 = 0$.

Hence,

$$\begin{aligned}
\boldsymbol{n}_t - \nabla F(\boldsymbol{x}_t) =& \beta(\boldsymbol{m}_t - \nabla F(\boldsymbol{x}_t)) + (1-\beta)(\boldsymbol{g}_t - \nabla F(\boldsymbol{x}_t)) \\
=& -\beta^{t+1} \nabla F(\boldsymbol{x}_1) + (1-\beta)\left( \sum_{k=1}^{t} \beta^{t-k+1}(\boldsymbol{g}_k - \nabla F(\boldsymbol{x}_k) + (\boldsymbol{g}_t - \nabla F(\boldsymbol{x}_t))\right) \\
&- \beta^2 \sum_{k=1}^{t} \beta^{t-k}(\nabla F(\boldsymbol{x}_k) - \nabla F(\boldsymbol{x}_{k-1})),
\end{aligned} \tag{31}$$

Then, we obtain

$$\begin{aligned}
\frac{1}{T} \sum_{t=1}^{T} \mathbb{E}\left[\|\boldsymbol{n}_t - \nabla F(\boldsymbol{x}_t)\|_2\right] \leq& \underbrace{\frac{\beta}{T} \sum_{t=1}^{T} \beta^t \|\nabla F(\boldsymbol{x}_1)\|_2}_{T_2} \\
&+ \underbrace{\frac{1-\beta}{T} \sum_{t=1}^{T} \mathbb{E}\left[\left\|\sum_{k=1}^{t} \beta^{t-k+1}(\boldsymbol{g}_k - \nabla F(\boldsymbol{x}_k))\right\|_2 + \|\boldsymbol{g}_t - \nabla F(\boldsymbol{x}_t)\|_2\right]}_{T_3} \\
&+ \underbrace{\frac{\beta^2}{T} \sum_{t=1}^{T} \mathbb{E}\left[\left\|\sum_{k=1}^{t} \beta^{t-k}(\nabla F(\boldsymbol{x}_k) - \nabla F(\boldsymbol{x}_{k-1}))\right\|_2\right]}_{T_4}
\end{aligned} \tag{32}$$

In terms of $T_2$, we obtain

$$T_2 = \frac{\beta}{T} \sum_{t=1}^{T} \beta^t \|\nabla F(\boldsymbol{x}_1)\|_2 \leq \frac{\beta}{(1-\beta)T} \|\nabla F(\boldsymbol{x}_1)\|_2 \tag{33}$$

As for $T_3$, we have

$$
T_3 = \frac{1-\beta}{T} \sum_{t=1}^{T} \mathbb{E} \left[ \left\| \sum_{k=1}^{t} \beta^{t-k+1}(\boldsymbol{g}_k - \nabla F(\boldsymbol{x}_k)) \right\|_2 + \|\boldsymbol{g}_t - \nabla F(\boldsymbol{x}_t)\|_2 \right]
$$

$$
\overset{(i)}{\leq} \frac{1-\beta}{T} \sum_{t=1}^{T} \sqrt{\mathbb{E} \left[ \left\| \sum_{k=1}^{t} \beta^{t-k+1}(\boldsymbol{g}_k - \nabla F(\boldsymbol{x}_k)) \right\|_2^2 + \|\boldsymbol{g}_t - \nabla F(\boldsymbol{x}_t)\|_2^2 \right]}
$$

$$
\overset{(ii)}{=} \frac{1-\beta}{T} \sum_{t=1}^{T} \sqrt{\sum_{k=1}^{t} \beta^{2(t-k+1)} \mathbb{E} \left[ \|\boldsymbol{g}_k - \nabla F(\boldsymbol{x}_k)\|_2^2 + \|\boldsymbol{g}_t - \nabla F(\boldsymbol{x}_t)\|_2 \right]} \qquad (34)
$$

$$
\overset{(iii)}{\leq} \frac{1-\beta}{T} \sum_{t=1}^{T} \sqrt{\sum_{k=1}^{t+1} \beta^{2(t-k+1)} \sigma^2}
$$

$$
\leq \frac{1-\beta}{\sqrt{1-\beta^2}} \sigma
$$

$$
\leq \sqrt{1-\beta} \sigma,
$$

where $(i)$ holds due to the fact $(\mathbb{E}[Z])^2 \leq \mathbb{E}[Z^2]$; $(ii)$ holds owing to $\mathbb{E}[\boldsymbol{g}_k - \nabla F(\boldsymbol{x}_k)] = \boldsymbol{0}$ according to Assumption 3; $(iii)$ holds resulting from $\mathbb{E}\left[ \|\boldsymbol{g}_k - \nabla F(\boldsymbol{x}_k)\|_2^2 \right] \leq \sigma^2$ according to Assumption 3.

Now we turn attention to $T_4$, *i.e.*,

$$
T_4 = \frac{\beta^2}{T} \sum_{t=1}^{T} \mathbb{E} \left[ \left\| \beta^{t-k}(\nabla F(\boldsymbol{x}_k) - \nabla F(\boldsymbol{x}_{k-1})) \right\|_2 \right]
$$

$$
\overset{(i)}{\leq} \frac{\beta^2}{T} \sum_{t=1}^{T} \sum_{k=1}^{t} \beta^{t-k} \mathbb{E} \left[ \|\nabla F(\boldsymbol{x}_k) - \nabla F(\boldsymbol{x}_{k-1})\|_2 \right]
$$

$$
\overset{(ii)}{\leq} \frac{\beta^2}{T} \sum_{t=1}^{T} \sum_{k=1}^{t} \beta^{t-k} \mathbb{E} \left[ (L_0 + L_1 \|\nabla F(\boldsymbol{x}_k)\|_2) \|\boldsymbol{x}_k - \boldsymbol{x}_{k-1}\|_2 \right]
$$

$$
\overset{(iii)}{=} \frac{\beta^2}{T} \sum_{t=1}^{T} \sum_{k=1}^{t} \beta^{t-k} \mathbb{E} \left[ \gamma (L_0 + L_1 \|\nabla F(\boldsymbol{x}_k)\|_2) \|\boldsymbol{u}_{t-1}\|_2 \right)
$$

$$
\overset{(iv)}{\leq} \frac{\beta^2}{T} \sum_{t=1}^{T} L_0 \gamma \sqrt{d} \sum_{k=1}^{t} \beta^{t-k} + \beta^2 L_1 \gamma \sqrt{d} \sum_{k=1}^{t} \beta^{t-k} \nabla F(\boldsymbol{x}_k) \qquad (35)
$$

$$
\leq \frac{\beta^2 L_0 \gamma \sqrt{d}}{1-\beta} + \beta^2 L_1 \gamma \sqrt{d} \sum_{t=1}^{T} \sum_{k=1}^{t} \beta^{t-k} \mathbb{E}[\|\nabla F(\boldsymbol{x}_k)\|_2]
$$

$$
\overset{(v)}{=} \frac{\beta^2 L_0 \gamma \sqrt{d}}{1-\beta} + \frac{\beta^2 L_1 \gamma \sqrt{d}}{T} \sum_{k=1}^{T} \mathbb{E}[\|\nabla F(\boldsymbol{x}_k)\|_2] \sum_{t=k}^{T} \beta^{t-k}
$$

$$
\leq \frac{\beta^2 L_0 \gamma \sqrt{d}}{1-\beta} + \frac{\beta^2 L_1 \gamma \sqrt{d}}{(1-\beta)T} \sum_{t=1}^{T} \mathbb{E}[\|\nabla F(\boldsymbol{x}_t)\|_2]
$$

$$
\overset{(vi)}{\leq} \frac{\beta^2 L_0 \gamma \sqrt{d}}{1-\beta} + \frac{\beta^2 L_1 \gamma \sqrt{d}}{(1-\beta)T} \sum_{t=1}^{T} \mathbb{E}[\|\nabla F(\boldsymbol{x}_t)\|_1]
$$

where $(i)$ holds due to the fact $\|\boldsymbol{a} + \boldsymbol{b}\|_2 \leq \|\boldsymbol{a}\|_2 + \|\boldsymbol{b}\|_2$; $(ii)$ holds owing to Assumption 2; $(iii)$ holds due to the update rule; $(iv)$ holds depends on $\boldsymbol{u}^{(j)} \leq 1$ according to Theorem 2; $(v)$ holds resulting from the fact that $\sum_{i=1}^{n} \sum_{j=1}^{i} \boldsymbol{a}_{i,j} = \sum_{j=1}^{n} \sum_{i=j}^{n} \boldsymbol{a}_{i,j}$; $(vi)$ holds due to the fact $\|\boldsymbol{a}\|_2 \leq \|\boldsymbol{a}\|_1$.

Combining Eq.(32) - Eq.(35), we have

$$
\begin{aligned}
\frac{1}{T}\sum_{t=1}^{T}\mathbb{E}\left[\|\boldsymbol{n}_t - \nabla F(\boldsymbol{x}_t)\|_2\right] \leq & \frac{\beta}{(1-\beta)T}\|\nabla F(\boldsymbol{x}_1)\|_2 + \sqrt{1-\beta}\sigma \\
& + \frac{\beta^2 L_0 \gamma\sqrt{d}}{1-\beta} + \frac{\beta^2 L_1 \gamma\sqrt{d}}{(1-\beta)T}\sum_{t=1}^{T}\mathbb{E}[\|\nabla F(\boldsymbol{x}_t)\|_1] \\
\leq & \frac{\beta}{(1-\beta)T}\|\nabla F(\boldsymbol{x}_1)\|_2 + \sqrt{1-\beta}\sigma \\
& + \frac{\beta^2 L_0 \gamma\sqrt{d}}{1-\beta} + \frac{\beta^2 L_1 \gamma\sqrt{d}}{(1-\beta)T}\sum_{t=1}^{T}\mathbb{E}[\|\nabla F(\boldsymbol{x}_t)\|_1]
\end{aligned}
\tag{36}
$$

Combining Eq.(28) and Eq.(36), we obtain

$$
\begin{aligned}
\frac{u_{\min} - \frac{\gamma d L_1}{2} - \frac{2\beta^2 L_1 \gamma\sqrt{d}}{1-\beta}}{T}\sum_{t=1}^{T}\mathbb{E}[\|\nabla F(\boldsymbol{x}_t)\|_1] \leq & \frac{F(\boldsymbol{x}_1) - F(\boldsymbol{x}^*)}{\gamma T} + \frac{2\beta\sqrt{d}}{(1-\beta)T}\mathbb{E}\left[\|\nabla F(\boldsymbol{x}_1)\|_2\right] \\
& + 2\sqrt{(1-\beta)d}\sigma + \frac{2\gamma\beta^2 L_0 d}{1-\beta} + \frac{\gamma d L_0}{2}.
\end{aligned}
\tag{37}
$$

Let $\gamma = \frac{1}{L_0 T^{3/4}}$, $1 - \beta = \frac{1}{T^{1/2}}$. When $T \geq \max\{(\frac{2dL_1}{L_0 u_{\min}})^{4/3}, (\frac{8\beta^2\sqrt{d}L_1}{(1-\beta)L_0 u_{\min}})^4\}$, we can guarantee

$$
u_{\min} - \frac{\gamma d L_1}{2} - \frac{2\gamma\sqrt{d}L_1}{1-\beta} \geq \frac{u_{\min}}{2}.
\tag{38}
$$

Then, setting $U_{\max} = \frac{1}{u_{\min}}$, we reformulate Eq. (37) as

$$
\begin{aligned}
\frac{1}{T}\sum_{t=1}^{T}\mathbb{E}[\|\nabla F(\boldsymbol{x}_t)\|_1] \leq & \frac{2L_0 U_{\max}(F(\boldsymbol{x}_1) - F(\boldsymbol{x}^*))}{T^{1/4}} + \frac{4\beta U_{\max}\sqrt{d}\mathbb{E}\left[\|\nabla F(\boldsymbol{x}_1)\|_2\right]}{T^{1/2}} \\
& + \frac{4U_{\max}\sqrt{d}\sigma}{T^{1/4}} + \frac{4\beta^2 U_{\max}d}{T^{1/4}} + \frac{U_{\max}d}{T^{3/4}}.
\end{aligned}
\tag{39}
$$

