# OpenReview forum: "SoftSignSGD(S3): An Enhanced Optimize for Practical DNN Training and Loss Spikes Minimization Beyond Adam"
_ICLR.cc/2025/Conference — Submitted to ICLR 2025_

### Official Review · Reviewer_RDxB · 2024-10-27

**Soundness:** 3
**Presentation:** 3
**Contribution:** 3
**Rating:** 6
**Confidence:** 2

**Summary:**

This paper proposes a novel optimizer, SignSoftSGD (S3), which uses higher-order moment in the denominator, uses the same momentum coefficient for numerator and denominator compared with Adam, and integrates Nesterov’s accelerated gradient technique.
These modifications are designed to address the training instability and loss spikes often encountered with Adam, while also enhancing performance. The authors provide a theoretical convergence analysis for S3 with O(1/T^{1/4}) convergence rate.
Finally, the auothers conduct extensive experiments on language and vision tasks to showcase the effectiveness and stability of S3.

**Strengths:**

- The authors propose an optimizer, S3, with theoretical convergence guarantee, primarily aimed at solving the training instability issues seen with Adam.
- The pretraining experiments on language and vision tasks are solid. Extensive ablation experiments clarify the contributions of each modification in S3 over Adam, with recommendations for a default setting of  \beta = 0.95  and  p = 3  in practice.

**Weaknesses:**

- My primary concern is regarding the cause of loss spikes in Adam.
	- Although Section 5.5 conducts several experiments to show Adam with same \beta successfully avoid loss spikes, the experiments in Section 5.3 display some inconsistent and intriguing behaviors.
	- In Figure 5, none of the S3 runs exhibit loss spike, even for configurations like (S3, lr=3e-3, diff. \beta, w/o NAG, p=3) and (S3, lr=3e-3, diff. \beta, w/ NAG, p=2). These results suggest that higher values of  p  and the inclusion of NAG may also prevent loss spikes. Therefore, it is better to provide a more detailed analysis of how each modification (higher p, NAG, same β) contributes to loss spike prevention.
	- Figure 5 only shows loss curve for part of the experiment setups in Table 3, thus it is not straightforward to see which part is the main factor of the loss spike. It is better to provide loss curves for all setups in Table 3.
	- Furthermore, in the GPT-2 pretraining tasks, the NAG incorporated optimizers including Adan, and NAdam both have loss spikes, which appears inconsistent with the findings in Figure 5.
	- I also suggest adding the standard deviation for all reported metrics.
- The bound in Theorem 4 depends on dimension d. Although removing such an dependence over d is technically hard, could you compare the dependence on d explicitly with other recent works on adaptive optimizers like Adam.
- What about the performance of S3 on diffusion models pretraining.
- Writing:
	- Please use  \citep and \citet formatting correctly.
	- There appear to be some typos or incomplete statements in the submission. For example, in line 161, it states, “An illustrative example can be found in Section D.3 of the appendix,” but there is no Appendix D.
	- In line 488, it shoud refer to Figure 6.

**Questions:**

See weakness above

---

> ### Author Response · Authors · 2024-11-19
> **Responses to RDxB （1/3）**
>
> We sincerely thank you for your voluntary efforts, constructive comments, and positive score. We are especially grateful for the note, *“The authors propose an optimizer, S3, with theoretical convergence guarantee, primarily aimed at solving the training instability issues seen with Adam.”* Furthermore, we are pleased that the solid and extensive experiments have been acknowledged. Below, we address your remaining concerns point-by-point.
>
> **Q1.** Clarifying the contributions of the modifications over Adam (*"NAG," "higher $p$," and "same $\beta$"*) to loss spike prevention.
>
> We greatly appreciate your keen observations and thoughtful analysis of our experimental results. Based on Theorems 1 and 2, the analyses in Section 2, and our experimental findings, we provide the following clarifications:
>
> 1. **NAG's impact:**
>    NAG has no direct connection to the reduction of loss spikes. According to Theorems 1 and 2, the maximum update magnitude remains the same regardless of whether NAG is applied, meaning it has minimal influence on loss spikes.
>
>    Regarding the setting *"lr=3e-3, diff. $\beta$, w/NAG, $p=2$"* in Figure 5, where no spikes are observed, the reasons are as follows:
>    - **Model Size:** The number of parameters in ViT-B is 86M, which is relatively small. The probability of loss spikes is positively correlated with model size, so spikes may not emerge with this random seed due to their low probability.
>    - **Recording Granularity:** During ViT-B training, we recorded training loss per epoch, whereas for GPT-2, loss was recorded per iteration. Consequently, some loss spikes might have occurred between epochs but were not captured.
>
>    To validate this, we retrained ViT-B/16 with the same configuration (*$lr=3e-3$, diff. $\beta$, w/NAG, $p=2$*) using a different random seed. The resulting curves, available at [this link](https://1drv.ms/b/s!Au3MrR-o69M4iMg5emYiYBkS0bfJqA?e=nDFnxn), show a loss spike, corroborating our explanation.
>
> 2. **Effect of "higher $p$":**
>    From Theorem 2, we know that:
>    $
>    \frac{\vert {n} _t^{(j)} \vert}{{b} _t^{(j)}(p_1)}({g}  _t^{(j)}, {g} _{t-1}^{(j)}, \dots) \geq \frac{\vert {n} _t^{(j)} \vert}{{b} _t^{(j)}(p_2)}({g} _t^{(j)}, {g} _{t-1}^{(j)}, \dots) \quad \text{if } p_1 \leq p_2.
>    $
>    This implies that higher $p$ reduces the maximum possible value of $\frac{\vert {n}_t^{(j)} \vert}{{b}_t^{(j)}(p)}$, thereby lowering the risk of loss spikes indirectly.
>
> 3. **Effect of "same $\beta$":**
>    As demonstrated in Theorem 2 and Section 3, "same $\beta$" directly minimizes the risk of loss spikes by controlling the update magnitude effectively.
>
> In summary, while NAG does not contribute to loss spike prevention, "higher $p$" indirectly reduces their probability by lowering the maximum upper bound, and "same $\beta$" directly minimizes the risk. These analyses will be further clarified in the revised manuscript to improve the reader's understanding.

---

> ### Author Response · Authors · 2024-11-19
> **Responses to RDxB （2/3）**
>
> **Q2.** *The bound in Theorem 4 depends on dimension $d$. Compare the dependence on $d$ explicitly with other recent works on adaptive optimizers like Adam.*
>
> **R2.** We summarize the convergence rates of S3 and Adam (from its original paper and subsequent recent works) as follows:
>
> 1. **The convergence rate of S3 (Our Work)**: From Theorem 4,
>    $
>    \frac{1}{T}\sum_{t=1}^T \mathbb{E}[\Vert\nabla F({x}_t)\Vert_1] \leq \mathcal{O}\left(\frac{d}{T^{1/4}}\right).
>    $
>
> 2. ** The convergence rate of Adam from (Kingma and Ba, 2015) [1]**:
>    Under convexity assumptions,
>    $
>    \frac{1}{T}\sum_{t=1}^T \left(f({x}_t) - f({x}^*)\right) \leq \mathcal{O}\left(\frac{d}{T^{1/2}}\right).
>    $
>
> 3. **The convergence rate of Adam in from (D{\'e}fossez et al., 2020) [2]**:
>    $
>    \frac{1}{T} \sum _{t=1}^T \mathbb{E}[\Vert\nabla F({x}_t)\Vert_2] \leq \mathcal{O}\left(\frac{d\ln(T)}{T^{1/4}}\right).
>    $
>
>    Since $\Vert a \Vert_1 \leq \sqrt{d}\Vert a \Vert_2$, the rate can be expressed as:
>    $
>    \frac{1}{T}\sum _{t=1}^T \mathbb{E}[\Vert\nabla F({x}_t)\Vert_1] \leq \mathcal{O}\left(\frac{d\sqrt{d}\ln(T)}{T^{1/4}}\right).
>    $
>
> 4. **The convergence rate of Adam from (Chen et al., 2022) [3]**:
>    $
>    \frac{1}{T}\sum _{t=1}^T \mathbb{E}[\Vert\nabla F({x}_t)\Vert_2] \leq \mathcal{O}\left(\frac{G\sqrt{d}\ln(T)}{T^{1/4}}\right),
>   $
>    where $G \ge \Vert {g}_t \Vert_2 \propto \sqrt{d}$. Thus,
>    $
>    \frac{1}{T}\sum _{t=1}^T \mathbb{E}[\Vert\nabla F({x}_t)\Vert_1] \leq \mathcal{O}\left(\frac{d\sqrt{d}\ln(T)}{T^{1/4}}\right).
>    $
>
> 5. **Adam (from Li et al., 2023) [4] In probabilistic bounds (Li el at. 2023)** ,
>    $
>    \frac{1}{T}\sum _{t=1}^T \Vert\nabla F({x}_t)\Vert_2 \leq \mathcal{O}\left(\frac{G^2\ln{\frac{1}{\delta}}}{T^{1/4}}\right), \text{ with probability at least } 1-\delta,
>    $
>    where $G \ge \Vert {g}_t \Vert_2 \propto \sqrt{d}$. Hence,
>    $
>    \frac{1}{T}\sum _{t=1}^T \mathbb{E}[\Vert\nabla F({x}_t)\Vert_1] \leq \mathcal{O}\left(\frac{d\sqrt{d}\ln{\frac{1}{\delta}}}{T^{1/4}}\right), \text{ with probability at least } 1-\delta.
>    $
>
> From this comparison, S3 may achieve a weaker dependency on \(d\) compared to Adam. We hope this clarification highlights the theoretical advantages of S3's convergence.
>
> **Q3.** *I also suggest adding the standard deviation for all reported metrics.*
>
> **R3.** Thank you for the valuable suggestion. We acknowledge that we only ran one trial per result due to computational constraints. Running multiple trials and reporting mean and standard deviation values would indeed make the results statistically more robust. However, like many in academia with limited computational resources, we have access to only two machines, each equipped with eight V100 GPUs. Each trial of training ViT-B/16 and ResNet-50 on ImageNet required approximately 20 hours, while training GPT-2 (345M) on OpenWebText took about 70 hours per trial. We spent over three months completing all experiments, running one trial per result, except for GPT-2 (7B), which required approximately 20 days on a rented cluster. Consequently, while we would have liked to conduct more experiments, our limited computational resources make this impractical.
>
> We emphasize that the datasets used, such as ImageNet and OpenWebText, are large-scale and diverse. This, coupled with results obtained from a single trial, provides reasonable confidence in their robustness. Moreover, the related Adan and Lion from industrial community also reported results based on single trials, indicating that this practice is acceptable for large-scale experiments.
> Notably, as for the most related Adan and Lion that come from the industrial community, they also only ran one trial for each results. In fact, the size of the datasets ImageNet and OpenWebText in our experiments is much huge, so the even one trial for each result will be still robust and convincing.
>
> **Q4.** *It is better to provide loss curves for all setups in Table 3.*
>
> **R4.** Thank you for the suggestion. We initially refrained from plotting all curves in the tables to avoid overlapping and cluttered legends. As suggested, we have included all curves in the revised manuscript. For convenience, the figure is also available at the following link https://1drv.ms/b/s!Au3MrR-o69M4iMgyf5FGMpdbJEd2cA?e=akqLug.

---

> > ### Author Response · Authors · 2024-11-19
> > **Responses to RDxB （3/3）**
> >
> > **Q5.** *What about the performance of S3 on diffusion models pretraining.*
> >
> > **R5.**  As suggested, we trained DDPM on CIFAR-10 using AdamW and S3. The training loss curves are available at the following link https://1drv.ms/b/s!Au3MrR-o69M4iMg-x7Bio4WVz8R1yg?e=hO4WhL. Experimental results indicate that S3 performs slightly better than Adam.
> >
> > **Q6.** *The typos and writing issues.*
> >
> > **R6.**  Thank you for this valuable reminder. We have thoroughly proofread the manuscript and corrected all typographical and writing issues in the revised version.

---

> ### Comment · Reviewer_RDxB · 2024-11-24
> **Official Comment by Reviewer RDxB**
>
> I sincerely thank the authors for their detailed responses to my comments and for conducting additional experiments. I greatly appreciate the effort and time invested. However, I still have some concerns regarding Q1, particularly in light of the figure provided in response to Q4.
>
> To effectively demonstrate that S3 can mitigate loss spikes, the experimental setup needs to be sufficiently robust. Unfortunately, I find the explanations provided by the authors not entirely convincing. Specifically:
>
> - The authors stated, *"The probability of loss spikes is positively correlated with model size, so spikes may not emerge with this random seed due to their low probability."* However, I would like to point out that loss spikes occur in **all** Adam runs, which makes the experimental results appear to be cherry-picked.
> - While I understand the limitations posed by resource constraints, if the randomness of the seed can have such a significant impact on experimental behavior, I strongly suggest conducting multiple trials to strengthen the conclusions.
> - The authors stated, *"Consequently, some loss spikes might have occurred between epochs but were not captured."* I believe this issue can be addressed through certain loss-averaging techniques.
> - To convincingly demonstrate that S3 can prevent loss spikes in Adam due to consistent \beta, it would be most effective to operate in a setting where **all runs with non-identical \beta** exhibit loss spikes, while **all runs with identical \beta** do not.
>
> Finally, regarding the diffusion model experiments: in the case of large-scale diffusion model pretraining, would loss spikes or training instability be a problem for Adam?
>
> Once again, I appreciate the authors' efforts and look forward to further clarifications.

---

> ### Author Response · Authors · 2024-11-25
> **Thanks to the Follow-up Feedbacks**
>
> Thank you for your thoughtful feedback and for providing us with the opportunity to further clarify our work. We sincerely appreciate your detailed comments and address your concerns as follows:
>
> ### **Regarding Loss Spikes**
>
> 1. **Experimental Setup and Seed Selection.**
>    In our experiments, including those conducted prior to your comments, we used a fixed random seed ("6666") for training ResNet-50 and ViT-B/16 on ImageNet. This is documented in the source code provided in the supplemental materials. The results where S3 (*lr=3e-3, diff. $\beta$, w/NAG, $p=2$*) did not exhibit a loss spike while Adam did were not selectively chosen. These results emerged naturally under this fixed seed.
>
>    The experiments presented in Section 5.3 and Figure 5 aimed to conduct ablation studies to evaluate the contributions of individual components to the overall performance. They were not intended to explore the underlying causes of loss spikes, which were specifically addressed in Section 5.5 and Figure 7. We clearly knew that training ViT-B/16 with Adam does not consistently lead to frequent loss spikes due to the model's size, making it less ideal for studying this phenomenon of loss spikes. For this reason, we did not mention loss spikes in Section 5.3, nor did we record per-iteration training loss to capture the detailed occurrences of loss spikes.
>
> 2. **Impact of Loss Spikes on Final Performance.**
>    Based on our observations, infrequent loss spikes during training have minimal impact on final performance. For example, as shown in Section 5.3, the test accuracy of AdamW, which encountered a loss spike, remains comparable to S3 (*lr=3e-3, same $\beta$, w/o NAG, $p=2$*) that did not exhibit spikes. Similarly, the results in Figure 3(c-d) and Table 1 show that the accuracy of AdamW after 300 epochs aligns with the reference values in TorchVision (https://pytorch.org/vision/main/models.html and https://github.com/pytorch/vision/tree/main/references/classification), which we cloned from GitHub as the basic source codes for the vision tasks, as described in Section B.1.
>
>     Given this robustness,the conclusions drawn from the presented results may remain valid without requiring multiple trials, especially considering the resource constraints that limit additional experiments at this time.
>
> 3. **Detailed Analysis of Loss Spikes.**
>    To verify the root causes of loss spikes, we fortunately had conducted detailed experiments in Section 5.5 using GPT-2 (345M),  including the scenarios you suggested (i.e., testing Adam with both identical and non-identical  $\beta_s$). The results demonstrate that:
>
>    - When $\beta_1$ and $\beta_2$ differ, loss spikes frequently occur in the early stages of training.
>    - When $\beta_1$ and $\beta_2$ are identical under otherwise identical conditions, the loss spikes are eliminated.
>    - Constraining $\max \left(\frac{|m_t^{(j)}|}{\sqrt{v_t^{(j)}}}\right) \leq 1$ while keeping $\beta_1 \neq \beta_2$ significantly reduces loss spikes.
>
>    These findings confirm that Adam with non-identical $\beta$ values can produce overlarge gradients, causing loss spikes, and setting $\beta_1 = \beta_2$ effectively mitigates this issue, as deeply analyzed in Sections 2 and 3.
>
> ---
>
> ### **Regarding Diffusion Model Experiments**
>
> While we have not conducted experiments on large-scale diffusion model pretraining with S3 or Adam using identical $\beta$, we believe that the proposed approach can mitigate training instabilities arising from loss spikes. Our algorithm is designed to be generalizable and is not limited to specific tasks or architectures.
>
>
> We hope these clarifications address your concerns comprehensively. Thank you once again for your valuable feedback, which has been instrumental in refining our work.

---

> ### Comment · Reviewer_RDxB · 2024-11-27
>
> Thank you for your response. Based on your reply, I will keep my score at 6 (marginal accept). However, if you can show experiments where S3, i.e. Adam with the same \beta, helps with convergence in a reasonable setting where loss spikes affect performance, I’m happy to raise my score to 8.
>
> Additionally, about my question on diffusion: I was asking if loss spikes are actually a problem when training diffusion models in practice since they also use the Adam optimizer.

---

> ### Author Response · Authors · 2024-11-28
> **Thank You for Your Continued Feedback**
>
> We sincerely appreciate your further comments and this opportunity to clarify them. Below, we provide detailed responses to your concerns.
>
>
> ### **Loss Spikes Affecting Convergence**
>
> Fortunately, Figure 7 in the manuscript visually illustrates that the frequent loss spikes caused by vanilla AdamW lead to a slight slowdown in convergence, while AdamW with the same $\beta$s helps restore the convergence rate. For your convenience, we provide Figure 7 at this link: https://1drv.ms/b/s!Au3MrR-o69M4iMVueg52QALbpzxStQ?e=b8lCTO .
>
> In this figure, we used AdamW under different conditions to train GPT-2 (345M) on OpenWebText. The experimental results show the following:
>
> - With vanilla AdamW and a learning rate of $1 \times 10^{-3}$, more frequent spikes occur compared to the baseline learning rate of $3 \times 10^{-4}$. This causes a slight slowdown in convergence and results in a higher final loss.
> - At a learning rate of $3 \times 10^{-3}$, even more frequent spikes emerge, eventually causing divergence.
> - When adopting AdamW with $\beta_1 = \beta_2$ and a learning rate of $1 \times 10^{-3}$, as predicted by our analysis in Section 3, the loss spikes disappear. This configuration achieves faster convergence and a lower final loss compared to vanilla AdamW with the baseline learning rate of $3 \times 10^{-4}$.
>
> In summary, when training DNNs encounters a loss spike, the loss often recovers in a short time. However, there is a small probability that loss will deviate from the intended trajectory due to the abrupt parameter changes,   as detailed in Section 2. When frequent loss spikes occur, the trajectory may derail intended trajectory completely with high probability, negatively affecting both the convergence rate and the final loss.
>
> This phenomenon is widely recognized by practitioners in the field of large language models (LLMs). To avoid the slowdown convergence and degraded performance from frequent loss spikes, practitioners would rather to resort to resource-wasting and time-wasting engineering strategies, *i.e.*,  skipping data batches before spikes occur or restarting training from nearby checkpoints [1][2]. Furthermore, some famous LLM groups have invested significant time and resources to study and mitigate this issue [3][4][5].
>
> References:
>
> [1] Aakanksha Chowdhery, et al., "Palm: Scaling language modeling with pathways," JMLR, 2023.
>
> [2] Igor Molybog, et al., "A theory on Adam instability in large-scale machine learning," arXiv:2304.09871, 2023.
>
> [3] Aohan Zeng, Xiao Liu, et al., "GLM-130B: An open bilingual pre-trained model," arXiv:2210.02414, 2022.
>
> [4] Hugo Touvron, et al., "Llama: Open and efficient foundation language models," arXiv:2302.13971, 2023.
>
> [5] Aiyuan Yang, et al., "Baichuan 2: Open large-scale language models," arXiv:2309.10305, 2023.
>
> ---
>
> ### **Loss Spikes in Training Diffusion Models**
>
> We have no direct experience in training large-scale diffusion models. However, our analysis of loss spikes caused by AdamW is generalizable. Therefore, if frequent loss spikes occur during the training of diffusion models, we think they could negatively affect convergence and final performance.
>
> ---
>
> We hope these clarifications address your concerns comprehensively. Sincerely thank you once again for your valuable feedback, which has been instrumental in refining our work.

---

> > ### Comment · Reviewer_RDxB · 2024-11-28
> >
> > Thank you for your quick response.
> >
> > I am not fully convinced by this experiment. In [2], it is stated: "Loss spikes are difficult to study because any reproduction of these spikes at a smaller scale is not necessarily caused by or remediated by the same factors as in larger scales." I am unclear whether the reasons behind the loss spikes in your experiments are consistent with those observed in Figure 1 of [4], Figure 3 of [5], or Figure 1 of [2].

---

> ### Author Response · Authors · 2024-11-28
>
> We are highly grateful for your prompt feedback. We would respectfully ask if you could specify which parts of the experiments in Figures 2 and 7 you found unconvincing. We believe that these experiments provide strong evidence supporting our detailed analysis in Section 2 on how Adam can potentially cause loss spikes.
>
> ---
>
> ### **Clarification about the Experimental Results**
>
> We would respectfully ask if you could specify which parts of the experiments in Figures 2 and 7 you found unconvincing. We believe that these experiments provide strong evidence supporting our detailed analysis in Section 2 on how Adam can potentially cause loss spikes.
>
> Before our work, the root cause of loss spikes was not well understood, leaving researchers and practitioners unable to reliably reproduce or prevent loss spikes during LLM training. Although [2] attempted to propose a theory for the phenomenon, they were unable to identify the fundamental cause. Their conclusion explicitly acknowledges this, stating:
> *"We conclude that at this point, there is no silver bullet to solve the problem, and the appropriate remedy depends on the specific setup of the large-scale training run."*
>
> In contrast, our work identifies the root cause of loss spikes and provides a simple method to address them. While the statement in [2], *"Loss spikes are difficult to study because any reproduction of these spikes at a smaller scale is not necessarily caused by or remediated by the same factors as in larger scales,"* may hold for their analysis, it does not apply to our findings.
>
> As noted by **Reviewer Xi7Z**, *"Loss spikes in LLM training is a long-standing problem, and this possible explanation and solution should be immediately useful for the academic community."* We believe that uncovering the root cause of loss spikes and providing a straightforward solution is indeed a significant contribution to the field.
>
> ---
>
> ### **Consistency with Observations in Related Work**
>
> Our analysis in Section 2 aligns well with the descriptions of loss spikes in [1][2][3][4][5]. For your convenience, we provide the following excerpt from our paper:
>
> *"Theorem 1 indicates that when \textsf{\small Adam} is employed, there exists a probability that the update of each element $\frac{\vert{m} _t^{(j)}\vert}{\sqrt{{v} _t^{(j)}}}$ can reach an excessively large value. For instance, with typical settings of $\beta_1=0.9$ and $\beta_2=0.999$, the update $\frac{\vert{m}_t^{(j)}\vert}{\sqrt{{v}_t^{(j)}}}$ could approach its theoretical maximum of $\frac{1-\beta_1}{\sqrt{1-\beta_2}\sqrt{1-\frac{\beta_1^2}{\beta_2}}}\simeq 7.27$, while the normal absolute value of the update is less than $1$. While the probability of any specific parameter's update reaching this maximal value is low, the probability that at least one parameter's update reaches this value is high due to the large number of parameters in large models (LLM).When a parameter's update is excessively large and the learning rate is also high, it is likely to deviate substantially from its intended trajectory. Such deviations may propagate to neighboring parameters through interconnections, triggering a chain reaction that culminates in loss spikes. This mechanism explains the frequent occurrence of loss spikes during LLM training, particularly in the initial stages when learning rates are higher. Specifically, the probability of loss spikes increases with the size of the LLM. In conclusion, vanilla Adam poses a significant risk of loss spikes during large-scale model training. Mitigating this problem requires strategies to constrain the maximum update magnitude for each parameter coordinate."*
>
> From this analysis, we can conclude:
>
> - Larger models experience more frequent loss spikes, a trend that is consistently recorded in [1][2][3][4]. This may explain why reports of training instability were rare before the era of LLMs.
> - Loss spikes are more likely to occur during the early stages of training, as also observed in [2][3][4][5].
> - The common shift in Adam settings for LLM training from the default $\beta_2 = 0.999$ to $\beta_2 = 0.95$ that is more close to $\beta_1=0.9$ can be explained by our analysis, as it helps mitigate the occurrence of loss spikes.
>
> Additionally, [2] notes that spikes requiring the batch-skipping trick may not appear in the figures of [2][4][5], which could affect the observations of such phenomena.
>
> ---
>
>
> We hope these clarifications address your concerns. Thank you once again for your enthusiastical continued feedbacks.

---

### Official Review · Reviewer_grTo · 2024-11-03

**Soundness:** 2
**Presentation:** 1
**Contribution:** 2
**Rating:** 5
**Confidence:** 4

**Summary:**

This paper is motivated by observations (as seen in previous works too) that the effectiveness of Adam for deep learning tasks is its SignSGD-like property rather than its dimension-wise adaptivity, and also, that the Adam's training instability and loss spikes can be attributed to overly large updates that may take place. To address this issue, the paper presents a new adaptive method called SoftSignSGD (S3) that incorporates three design choices: p-th order moments, same $\beta$, and Nesterov's acceleration. The paper presents theoretical analyses on update scales as well as convergence properties. S3 is evaluated for vision and language tasks, achieving improved results in terms of task performances as well as training speed.

**Strengths:**

Recent works suggest that the SignSGD is what makes Adam and other adaptive methods work, and this work takes a step further to incorporate algorithmic advances so as to improve training stability and numerical effectiveness. The authors evaluate the proposed method (S3) for both vision and language tasks, and based on the results it indeed appears to be achieving superior performances for standard deep learning tasks.

**Weaknesses:**

- The assumption under which Theorem 1 holds (the signs of gradients are the same..) seems quite strong and may not hold in general. This makes the argument (rephrased) "large changes in specific and neighboring dimensions, leading to a chain reaction that ultimately results in loss spikes” rather speculative based on ad-hoc intuition (although Figure 2 seems still valid).
- Theorem 4 still has dependency on the problem dimensionality `d`.
- The claimed advantage of flexible p doesn’t seem to render much value; there doesn’t seem to exist a clear pattern of which p yields the best result, and thus, it anyway needs to be selected with a hyperparameter search. The criticism toward other compared methods due to their need of hyperparameter tuning thus seem a bit unfair or in other words, the contribution of allowing higher/flexible p seems a bit inflated.
- The improvement made by including Nesterov acceleration seems quite obvious (and also random) rather than fitting the purpose of this work. Also, it is hard to be considered their contribution unless the authors show that other compared methods cannot be improved with a similar acceleration scheme.
- The numerical results may not be statistically robust unless provided with results over multiple runs.
- This paper needs a lot of proof-reading; there are too many typos.

**Questions:**

- How many runs are these results averaged over? Is it the result of 1 (or n) run(s) using the same seed across all experiments? Can authors comment on this with respect to Tables 1, 2, and 3?

---

> ### Author Response · Authors · 2024-11-19
> **Responses to Reviewer grTo (1/2)**
>
> We sincerely thank you for your voluntary efforts and constructive comments. We appreciate your acknowledgment of the significance of minimizing the risk of loss spikes via soft-sign descent and the solid experimental evidence supporting our approach. Below, we address your remaining concerns point by point.
>
> **Q1.** *The assumption under which Theorem 1 holds (the signs of gradients are the same.) seems quite strong and may not hold in general.*
>
> **R1.**
> Thank you for pointing this out. We agree that the assumption is stringent and may not always hold strictly. However:
> 1. The update $\frac{\vert{m} _t^{(j)}\vert}{\sqrt{{v}_t^{(j)}}}$ with respect to $({g} _k^{(j)}) _{k=1}^t$ is a continuous function. Thus, when most of the signs of $({g} _k^{(j)}) _{k=1}^t$ are consistent and the secondary condition is nearly satisfied, $\frac{\vert{m} _t^{(j)}\vert}{\sqrt{{v} _t^{(j)}}}$ will be close to the theoretical maximum.
> 2. Historical gradients (${g} _k^{(j)}$) from much earlier iterations have minimal influence on the current value of $\frac{{m} _t^{(j)}}{\sqrt{{v} _t^{(j)}}}$ due to the moving exponential average.
>
> As a result, it becomes easier for $\frac{\vert{m} _t^{(j)}\vert}{\sqrt{{v}_t^{(j)}}}$  to reach a relatively high value, probabilistically leading to loss spikes. This is supported by Figure 2, which shows that the peaks of $\max _{j \in [d]} \left( \frac{\vert {m} _t^{(j)} \vert}{\sqrt{{v} _t^{(j)}}} \right)$ during a loss spike are close to, but not strictly equivalent to, the theoretical maximum of 7.27.
>
> In summary, while the assumption in Theorem 1 may not hold strictly, it provides a strong theoretical basis for explaining loss spikes when using Adam. We will incorporate this discussion into the revised manuscript. Your feedback has greatly improved the clarity of the paper.
>
>
>  **Q2.** *Theorem 4 still has a dependency on the problem dimensionality $d$.*
>
> **R2.**
> You are correct that the convergence bound of S3 in Theorem 4 depends on the dimensionality \(d\). However, its dependency is weaker than Adam's, as demonstrated in recent works. Below, we summarize the convergence rates for S3 and Adam for comparison:
>
> 1. **The convergence rate of S3 (Our Work)**: From Theorem 4,
>    $
>    \frac{1}{T}\sum_{t=1}^T \mathbb{E}[\Vert\nabla F({x}_t)\Vert_1] \leq \mathcal{O}\left(\frac{d}{T^{1/4}}\right).
>    $
>
> 2. ** The convergence rate of Adam from (Kingma and Ba, 2015) [1]**:
>    Under convexity assumptions,
>    $
>    \frac{1}{T}\sum_{t=1}^T \left(f({x}_t) - f({x}^*)\right) \leq \mathcal{O}\left(\frac{d}{T^{1/2}}\right).
>    $
>
> 3. **The convergence rate of Adam in from (D{\'e}fossez et al., 2020) [2]**:
>    $
>    \frac{1}{T} \sum _{t=1}^T \mathbb{E}[\Vert\nabla F({x}_t)\Vert_2] \leq \mathcal{O}\left(\frac{d\ln(T)}{T^{1/4}}\right).
>    $
>
>    Since $\Vert a \Vert_1 \leq \sqrt{d}\Vert a \Vert_2$, the rate can be expressed as:
>    $
>    \frac{1}{T}\sum _{t=1}^T \mathbb{E}[\Vert\nabla F({x}_t)\Vert_1] \leq \mathcal{O}\left(\frac{d\sqrt{d}\ln(T)}{T^{1/4}}\right).
>    $
>
> 4. **The convergence rate of Adam from (Chen et al., 2022) [3]**:
>    $
>    \frac{1}{T}\sum _{t=1}^T \mathbb{E}[\Vert\nabla F({x}_t)\Vert_2] \leq \mathcal{O}\left(\frac{G\sqrt{d}\ln(T)}{T^{1/4}}\right),
>   $
>    where $G \ge \Vert {g}_t \Vert_2 \propto \sqrt{d}$. Thus,
>    $
>    \frac{1}{T}\sum _{t=1}^T \mathbb{E}[\Vert\nabla F({x}_t)\Vert_1] \leq \mathcal{O}\left(\frac{d\sqrt{d}\ln(T)}{T^{1/4}}\right).
>    $
>
> 5. **Adam (from Li et al., 2023) [4] In probabilistic bounds (Li el at. 2023)** ,
>    $
>    \frac{1}{T}\sum _{t=1}^T \Vert\nabla F({x}_t)\Vert_2 \leq \mathcal{O}\left(\frac{G^2\ln{\frac{1}{\delta}}}{T^{1/4}}\right), \text{ with probability at least } 1-\delta,
>    $
>    where $G \ge \Vert {g}_t \Vert_2 \propto \sqrt{d}$. Hence,
>    $
>    \frac{1}{T}\sum _{t=1}^T \mathbb{E}[\Vert\nabla F({x}_t)\Vert_1] \leq \mathcal{O}\left(\frac{d\sqrt{d}\ln{\frac{1}{\delta}}}{T^{1/4}}\right), \text{ with probability at least } 1-\delta.
>    $
>
> From this comparison, S3 may achieve a weaker dependency on \(d\) compared to Adam. We hope this clarification highlights the theoretical advantages of S3's convergence.
>
>
>
>  [1] Diederik Kingma and Jimmy Ba. "Adam: A method for stochastic optimization". ICLR, 2015.
>
>  [2] Alexandre D{\'e}fossez, L{\'e}on Bottou, Francis Bach, and Nicolas Usunier. "A simple convergence proof of adam and adagrad". arXiv:2003.02395, 2020.
>
>  [3] Congliang Chen, Li Shen, Fangyu Zou, and Wei Liu. "Towards practical adam: Non-convexity, convergence theory, and mini-batch acceleration. JMLR, 2022.
>
>  [4] Haochuan Li, Alexander Rakhlin, and Ali Jadbabaie. "Convergence of adam under relaxed assumptions". NeurIPS, 2023.

---

> > ### Author Response · Authors · 2024-11-19
> > **Responses to Reviewer grTo (2/2)**
> >
> > **Q3.** *The claimed advantage of flexible $p$ doesn’t seem to render much value;  there doesn’t seem to exist a clear pattern of which p yields the best result.*
> >
> > **R3.** We respectfully disagree with you on this issue. As you mentioned in the strengths , one of our novelties is that we revolute the design ethos of an effective adaptive optimizer from the perspective of the sign-like descent according to recent findings rather than the simplification of the second-order optimizer from the original paper on Adam. As a result, we are the first to introduce a more flexible $p$-order momentum in the denominator, not limited to $2$-order momentum. We think the introduction of $p$-order momentum, thinking out of the box, may give inspiration for others to design new adaptive optimizers, which will bring more value to the community.
> >
> > Moreover, as shown in Figure 6, when $p=3$ commonly achieve the best or closely best performance over different schemes. Thus, we think setting $p=3$ as the default may be a relative good choice if we do not have time or resources to tune this hyperparameter.  In fact, when training GPT-2, we directly set  $p=3$ without tuning, it indeed achieved a superior performance, compared to the baseline Adam.
> >
> > Additionally, it is a little strange why it is unfair that the gain from the flexible $p$-order momentum in S3 is unfair, since the introduction of the flexible $p$-order momentum is just one of our novelties. On the other hand, as shown in Figure 6 and Table 4 in the appendix, even we also fix $p=2$ like the baseline Adam, S3 still outperforms Adam with a large gap.
> >
> >
> > **Q4.** *It is hard to consider the intergradation of NAG as the contribution since NAG is not the first introduced.*
> >
> > **R4.** As we repeatedly emphasize in our manuscript, we are not the first to introduce NAG to adaptive optimizers, but our new contribution is that we skillfully integrate NAG with a new formulation to the soft-sign descent without incurring extra memory costs for the practical purpose, so that S3 can enjoy fast training acceleration and better performance without the troubles of loss spikes and memory limitation, which make it more appropriate to train large models, such as LLMs, compared to the pioneering NAdam and Adan.
> >
> >
> > **Q5.** *The numerical results may not be statistically robust unless provided with results over multiple runs.*
> >
> > **R5.** Sincerely thank you for this valuable feedbacks. We admit that we only ran 1 trial for each result. We understand that running many trials for each results and reporting the mean value and the standard derivation will make the results more robust. However, like many people who in academia have limited computation resources, we only have two machines equipped 8 GPUs (V100). Each trial of training ViT-B/16 and ResNet-50 on ImageNet with one machine took about 20 hours, and each trial of training GPT-2 (345M) on OpenWebText with one machine took about 70 hours. We spent more than 3 months to complete all the experiments with running 1 trial for each result, except that we run GPT-2 (7B) for about 20 days in a cluster charged by hour. Consequently, it may be unrealistic for us to run many trials for each results.
> >
> > On the other hand, we would like to emphasize that the size of the datasets ImageNet and OpenWebText in the experiments is much huge, so even one trial for each result may be still robust and convincing enough. In fact,  as for the most related Adan and Lion that come from the industrial community, they also only ran one trial for each results, and they also thought one trial for each trial on large distastes is robust enough.
> >
> >
> > **Q6.** *This paper needs a lot of proof-reading; there are too many typos.*
> >
> > **R6.** Thank you sincerely for this kind reminder. We have carefully proofread the manuscript and corrected all errors and writing issues in revised version.

---

> ### Author Response · Authors · 2024-11-26
>
> We sincerely appreciate your valuable feedback and constructive suggestions, which have significantly improved the quality and presentation of our work. We kindly ask whether our response has addressed your concerns, and we would be happy to provide further clarifications or address any additional questions to ensure a clear understanding of our contributions. If you find our response satisfactory, we would be grateful if you would consider raising your rating.
>
>  We look forward to your reply.

---

> ### Author Response · Authors · 2024-12-02
> **Reviewer-Author Discussion Deadline Approaching**
>
> We sincerely appreciate the time and effort you have dedicated to reviewing our work. As the deadline for the reviewer-author discussion is approaching, we kindly ask if you could confirm whether our response has addressed your original concerns. If you are satisfied with the response, we would be grateful if you would consider updating your rating.
>
> Thank you once again for your valuable feedback and for your time.

---

### Official Review · Reviewer_Y3k2 · 2024-11-03

**Soundness:** 3
**Presentation:** 3
**Contribution:** 3
**Rating:** 6
**Confidence:** 3

**Summary:**

This paper investigates Adam's strengths and limitations, revealing that its success stems from a sign-like approach rather than second-order optimization. To address Adam's susceptibility to loss spikes, the authors propose SignSoftSGD (S3), an optimizer with a flexible momentum term and integrated Nesterov acceleration. S3 enhances training stability, reduces tuning needs, and shows superior performance across tasks with faster convergence and greater stability, even at larger learning rates. The authors' theoretical insights and practical results suggest that S3 could be especially beneficial in large language model training.

**Strengths:**

1. The paper is clearly written and easy to follow.
2. The issue of loss spikes in LLMs is a very interesting question and warrants further investigation.
3. The paper provides valuable theoretical insights into the convergence of the S3 algorithm.
4. The paper includes a detailed empirical investigation, including experiments and an ablation study.

**Weaknesses:**

1. The paper "On the Variance of the Adaptive Learning Rate and Beyond" suggests that the variance in adaptive learning rates could be the reason that causes training instability in the initial stages, which I believe is highly correlated to the authors' insights in this work. Could the authors discuss this connection and, if possible, provide demonstrations to highlight how the variance of adaptive learning rates changes when using S3? Such an analysis would strengthen the understanding of why training instability happens.

2. The authors argue that large m/sqrt(v) values may lead to the emergence of loss spikes, which I agree. Could the authors consider manually setting a large m/sqrt(v) in specific coordinates to intentionally trigger a loss spike? This would provide compelling evidence to support the claim and enhance the understanding of the mechanism behind these spikes.

3. The authors claim that "S3 employs the same exponential moving average coefficient β for both mt and rt, offering the advantages of minimizing the risk of loss spikes and reducing tuning work. In theory, as demonstrated in Theorem 2, the same β guarantees that the largest value of each coordinate of the update bjtn(tpj) is minimized to 1." However, I am not convinced that reducing one hyperparameter is inherently advantageous. In fact, we could also control the bound by carefully setting a relationship between beta_1 and beta_2 in Adam. Keeping one more parameter could give on more freedom of tuning, allowing to control the decay speed of higher-order gradient statistics.

4. Adaptive optimizers like Adam often require stable gradient accumulation in the early stages of training, and warmup techniques are commonly used to facilitate this stability. However, it seems that the authors have not mentioned using a warmup technique. Considering the loss spikes are often occurred at the beginning of training, could the authors clarify the specific cases in which they applied or chose not to apply warmup?

5. Some typos: line 205 "a excessive" -> "an excessive", line 231 "the study itself the study itself" -> "the study itself", Line 335 "Algorithm 4" -> "Theorem 4". Additionally, some aspects of the typesetting are inappropriate and need improvement, such as Line 344 - Line 345. The related work section should be in the main text.

6. It would be helpful if the authors could release the code.

**Questions:**

See Weakness

---

> ### Author Response · Authors · 2024-11-19
> **Responses to  Y3k2 (1/2)**
>
> We sincerely appreciate your voluntary efforts and positive feedback.We are particularly grateful for your observation: "The issue of loss spikes in LLMs is a very interesting question and warrants further investigation." Additionally, we are encouraged by the recognition of our theoretical analyses for S3 and the acknowledgment of our empirical experiments.  We address your remaining concerns point-by-point below.
>
>
> **Q1.** *The connection with the paper "On the Variance of the Adaptive Learning Rate and Beyond" [1].*
>
> **R1.**  Your valuable insights help us establish a strong connection between our work and [1]. Although we had carefully studied [1] earlier, we did not recognize this connection.  [1] empirically and theoretically demonstrated that the variance of the update $\frac{1}{\sqrt{{v}_t^{(j)}}}$ for each coordinate is significantly larger in the early stages, often causing $\frac{\vert{m}_t^{(j)}\vert}{\sqrt{{v}_t^{(j)}}}$ to become disproportionately large.  Consequently,  according to our analyses in Section 2, the train loss will encounter  frequent loss spikes if the learning rate is also large, finally leading to train instability.  Similarly, Theorem 1 in our work suggests that $\frac{\vert{m}_t^{(j)}\vert}{\sqrt{{v}_t^{(j)}}}$ is likely to become disproportionately large in the early stages, as conditions such as consistent sign patterns in $({g} _t^{(j)}, {g} _{t-1}^{(j)}, ...)$ and geometric decay of gradients are more easily satisfied at this stage. Furthermore, the analyses in both [1] and our work explain why the warmup technique enhances training stability during the early stages.
>
> Thank you for this insightful observation, which has deepened our understanding of early-stage training stability without warmup. We have incorporated this finding into the revised manuscript.
>
>
> **Q2.** *Could the authors consider manually setting a large $\frac{{m} _t^{(j)}}{\sqrt{{v} _t^{(j)}}}$ in AdamW in specific coordinates to intentionally trigger a loss spike?*
>
> **R2.** Thank you for this insightful suggestion. Following your suggestion, we conducted an experiment training a 6-layer ViT using AdamW on CIFAR-10. Due to the relatively small number of parameters in the 6-layer ViT, the probability of naturally occurring training loss spikes is low, as discussed in Section 2. We randomly selected parameters in the third layer every 1000 iterations and manually set their updates $\frac{{m} _t^{(j)}}{\sqrt{{v} _t^{(j)}}}$ to $20$.  The training loss curves are available at the following link: [provide link].  The loss curves exhibit some spikes, validating your hypothesis.  Notably, loss spikes do not occur every 1000 iterations because abrupt changes in individual parameters do not always impact the overall loss. 	In summary, this suggestion has significantly deepened our understanding of the mechanisms behind loss spikes.
>
> Furthermore, in addition to Figure 2 in Section 2, we conducted additional experiments in Section 5.5 to further investigate the causes of loss spikes from Adam, providing additional evidence and insights.
>
> *Q3.** *Clarifying the specific cases in which they applied or chose not to apply warmup.*
>
> **R3.** Due to space constraints in the main text, we moved the detailed experimental settings, including warmup configurations, to Section B.1 of the appendix, which may have been overlooked.  n Section B.1, we specified that the same warmup technique was applied across all experiments for the optimizers.  For vision tasks, we linearly increased the learning rate to its peak over the first 30 warmup epochs, followed by a cosine decay to zero in subsequent epochs. For language tasks, we linearly increased the learning rate to its peak over the first 5000 warmup steps, followed by a cosine decay to $0.1\times$ of the peak in subsequent steps.

---

> > ### Author Response · Authors · 2024-11-19
> > **Responses to Y3k2 (2/2)**
> >
> > **Q4.** *Using different $\beta_1$ and $\beta_2$ for the  numerator  momentum  ${m}_t$ and the denominator momentum ${v}_t$ will give more freedom of tuning, and the bound of the update $\frac{{m}_t}{\sqrt{{v}_t}}$ can be controlled by carefully setting a relationship between $\beta_1$ and $\beta_2$ in Adam.*
> >
> > **Q4.** You raise an interesting point; however, we respectfully hold a different perspective. There is no conclusive evidence from published literature or our experiments suggesting that using different $\beta$s to control the decay speed of higher-order gradient statistics significantly improves performance, even without loss spikes. In contrast, adding a tunable hyperparameter would substantially increase the number of trials required to find optimal combinations through grid search. Specifically, Specifically, even a single trial for training LLMs incurs significant time and monetary costs.  Moreover, as extensively discussed in Section 2, using different $\beta$s increases the risk of loss spikes in high-dimensional parameter spaces, potentially degrading performance.  AAdditionally, we have theoretically demonstrated that when $\beta_1$ and $\beta_2$ differ in Adam, the upper bound of $\frac{{m}_t^{(j)}}{\sqrt{{v}_t^{(j)}}}$ is inevitably higher than when $\beta_1$ and $\beta_2$ are the same, regardless of careful tuning.
> >
> >
> > **Q5.** *The typos and writhing issues.*
> >
> > **R5.** Thank you sincerely for this kind reminder. We have carefully proofread the manuscript and corrected all errors and writing issues in the revised version.
> >
> > **Q6.** *It would be helpful if the authors could release the code.*
> >
> > **R6.** As suggested, we have included the source code in the supplementary materials and will publicly release it as  paper is accepted

---

> > > ### Author Response · Authors · 2024-11-26
> > >
> > > We sincerely appreciate your valuable feedback and constructive suggestions, which have significantly improved the quality and presentation of our work. We kindly ask whether our response has addressed your concerns, and we would be happy to provide further clarifications or address any additional questions. We look forward to your reply.

---

> > > > ### Comment · Reviewer_Y3k2 · 2024-11-26
> > > >
> > > > Thank you for your responses! My concerns have been addressed and I would like to keep my original scores.

---

> > > > > ### Author Response · Authors · 2024-11-26
> > > > >
> > > > > We sincerely thank you for your positive reply and are pleased to hear that all concerns have been addressed. Given this, we respectfully ask whether it might be possible to raise the rating or at least increase the confidence level.
> > > > >
> > > > > Thank you again for the time and effort you have invested in evaluating our work.

---

### Official Review · Reviewer_CGUg · 2024-11-04

**Soundness:** 3
**Presentation:** 3
**Contribution:** 3
**Rating:** 6
**Confidence:** 3

**Summary:**

This paper reveals that the effectiveness of Adam in training complicated DNNs stems primarily from its similarity to SignSGD in managing significant gradient variations. The authors theoretically and empirically demonstrate that Adam is the underlying factor causing loss spikes in training large models (i.e., LLM) due to its potential to lead some parameter updates to be excessively large. Meanwhile, the authors introduce a novel optimizer, named SoftSignSGD (S3), which offers multiple distinct advantages over Adam. Finally, this paper provide theoretical analysis and extensive experiments to evaluate S3.

**Strengths:**

1) The paper proposes a novel optimizer, SignSoftSGD (S3), which gives smaller training loss and better test performance, and this is validated by a wide range of experiments.

2) This paper provide theoretical analysis for S3 on a general nonconvex stochastic problem and demonstrate that S3 achieves the optimal convergence rate under some assumption. This gives more evidence on the superior of the S3 optimizer.

**Weaknesses:**

It would be better if more experimental results are provided.

**Questions:**

1) Is the S3 optimizer sensitive to some parameters, such as the learning rate?
2) Is there any drawbacks of the S3 optimizer?

---

> ### Author Response · Authors · 2024-11-19
> **Responses to CGUg**
>
> We sincerely thank you for your voluntary efforts and positive feedback. We are pleased to hear that you found the experimental results supportive. More importantly, we are encouraged by your recognition of the theoretical significance of S3's convergence analysis. We address your remaining concerns point-by-point below.
>
>
> **Q1.** *Is the S3 optimizer sensitive to some parameters, such as the learning rate?*
>
> **R1.** Thank you for highlighting this important detail. While the learning rate plays a significant role in the performance of all optimizers, Figure 8 in the appendix shows that S3 is not more sensitive to the learning rate than other optimizers. We also conducted sensitivity analyses for $\beta$ and
> $p$ of S3 in Section 5.4. The results demonstrate that S3 is not highly sensitive to $\beta$ or $p$; even the worst-performing parameter combination outperforms baseline AdamW.
>
>
> **Q2.** *Are there any drawbacks of the S3 optimizer?*
>
> **R2.** The value of $p$ in S3 can be any value greater than or equal to 1. However, setting $p$ as a floating-point number rather than an integer increases computational cost, compared to the baseline Adam.
>
> **Q3.** *It would be better if more experimental results are provided.*
>
> **R3.**  Thank you for this valuable feedback. We understand that additional experiments would make the results even more compelling. However, like many in academia with limited computational resources, we have access to only two machines, each equipped with eight V100 GPUs. Each trial of training ViT-B/16 and ResNet-50 on ImageNet required approximately 20 hours, while training GPT-2 (345M) on OpenWebText took about 70 hours per trial. We spent over three months completing all experiments, running one trial per result, except for GPT-2 (7B), which required approximately 20 days on a rented cluster. Consequently, while we would have liked to conduct more experiments, our limited computational resources make this impractical. Nevertheless, our experiments cover widely-used vision and language tasks, including popular CNN and Transformer architectures. These results, acknowledged as solid by **Reviewer Xi7Z** and **Reviewer RDxB**, provide extensive validation.

---

> > ### Author Response · Authors · 2024-11-26
> >
> > We sincerely appreciate your valuable feedback and constructive suggestions, which have significantly improved the quality and presentation of our work. We kindly ask whether our response has addressed your concerns, and we would be happy to provide further clarifications or address any additional questions. We look forward to your reply.

---

### Official Review · Reviewer_Xi7Z · 2024-11-04

**Soundness:** 4
**Presentation:** 4
**Contribution:** 4
**Rating:** 8
**Confidence:** 3

**Summary:**

Authors present SoftSignSGD (S3), an optimizer designed to address the problem of Adam optimizer, which may cause loss spikes. S3 uses a p-order momentum which resembles the behavior of SignSGD. Authors provide strong results on ResNet/ViT on ImageNet and OpenWebText on GPT-2.

**Strengths:**

1. The visualization in figure 2 is insightful where authors build a connection between mean update and train loss spikes.
2. The paper is nicely motivated.
3. Experiments are solid and results are strong.
4. Loss spikes in LLM training is a long-standing problem, and this possible explanation and solution should be immediately useful for the academic community.

**Weaknesses:**

See questions

**Questions:**

1. If p is not equal to 1, how will the wall clock time per iteration change? (compared to Adam and S3 p=1)
2. How is the learning rate for different optimizers determined? Are they properly tuned for baselines?
3. Will tuning betas for Adam (and getting a smoother loss curve) close the gap between Adam and S3?

---

> ### Author Response · Authors · 2024-11-19
>
> We sincerely thank you for your voluntary efforts and high recognition. More importantly, we are gratified that our explanation of loss spikes originating from vanilla Adam is recognized and that our proposed S3, which minimizes the risk of loss spikes, is considered an immediate contribution to the academic community.  We address your concerns point-by-point below. following.
>
> **Q1.** *If $p$ is not equal to 1, how will the wall clock time per iteration change? (compared to Adam and S3 $p=1$)*
>
> **R1.** As suggested, we have re-trained GPT-2(345M) with S3 (p=1), S3 (p=2), and AdamW for 1000 iterations to record the wall clock time. The results are summarized below.
>
> *Table 1. The mean time per iteration for training  GPT-2(345M) with S3 ($p=1$), S3 ($p=3$)  and AdamW*
> |                         |   S3 ($p=1$) | S3 ($p=3$) | AdamW      |
> |:-----------             | :-----------:|:-----------:|:---------:|
> |Mean Time per iter (ms)  |  4623.2      | 4703.4      | 4659.3   |
>
>
> As shown in Table 1, S3 ($p=3$) costs slightly higher time per iteration than AdamW, while S3 ($p=1$) is slightly faster than AdamW.
>
>
> **Q2.** *How is the learning rate for different optimizers determined? Are they properly tuned for baselines?*
>
> **R2.** Highly thank you for taking attention to this important detail. Due to space constraints, we provided the detailed settings of the learning rates for the baseline optimizers in Section B.1 of the appendix. A brief summary is provided here for your convenience.
>
> During training for ResNet-50 and ViT-B/16, learning rates for all optimizers are linearly increased to their peak over the first 30 epochs, followed by a cosine decay to 0 in subsequent epochs. The peak learning rates are as follows:
>
>  + For SGD, we set $lr_{\max}=0.3$ , the default value in the official PyTorch vision reference codes.
>
>  + For AdamW, we utilize $lr_{\max}=0.003$ , the default value in the official PyTorch vision reference codes and also widely adopted in related studies [1][2].
>
>  + For Adan, we employe $lr_{\max}=0.015$, following official recommendations[3].
>
>  + For Lion, we adopt $lr_{\max}=0.001$, following official recommendations [2].
>
>  For GPT-2 (345M), we linearly increase learning rates for  all the optimizers to the peak in the initial $5k$ steps and then decrease to $0.1\times$ of the peak with a cosine decay in the subsequent steps. Peak learning rates for all optimizers were determined through coarse search while training GPT-2 (345M) (see Figure 8 in the appendix). For GPT-2 (7B), the learning rates for AdamW followed settings from the LLaMA paper [4].
>
> [1] J. Zhuang, et al. "AdaBelief optimizer: Adapting stepsizes by the belief in observed gradients". NeurIPS, 2020.
>
> [2] X. Chen, et al. "Symbolic discovery of optimization algorithms". arXiv:2302.06675, 2023.
>
> [3] X. Xie, et al. "Adan: Adaptive nesterov momentum algorithm for faster optimizing deep models". TPAMI, 2024.
>
> [4] H. Touvron, et al. "Llama: Open and efficient foundation language models".  arXiv:2302.13971, 2023.
>
>
> **Q3.**  *Will tuning betas for Adam (and getting a smoother loss curve) close the gap between Adam and S3?*
>
> **R3.** Thank you for the thoughtful suggestion. As we currently have access to only one machine with 8 V100 GPUs, it is not feasible to perform an exhaustive grid search over $\beta_1$
> and $\beta_2$ for AdamW within the short response period. We conducted experiments using AdamW,  fixing $\beta_1=0.9$ and changing $\beta_2$ to train ViT-B/16 on ImageNet. The results are summarized below:
>
> Table 2. Impact of $\beta_1$ and $\beta_2$ on the Accuracy of AdamW training ViT-B/16 on ImageNet
> |         |  $\beta_2=0.9$  |   $\beta_2=0.95$  |   $\beta_2=0.99$  | $\beta_2=0.999$  |
> |:--------|:---------------|:---------------|:---------------|:---------------|
> |$\beta_1=0.9$|    79.45       |      79.48        |        79.43   |         79.52     |
>
> As shown in Table 2, tuning $\beta$s may not enable Adam to achieve performance comparable to  S3.

---

> > ### Comment · Reviewer_Xi7Z · 2024-11-25
> >
> > Thanks authors for the additional experiments and information.
> >
> > ADAM optimizer's beta has a large effect on the smoothness of training curve, I encourage authors to look further into this matter after the rebuttal period.
> >
> > Overall I feel this work is solid and will keep my original positive score.

---

> > > ### Author Response · Authors · 2024-11-26
> > >
> > > We sincerely thank you for your positive rating and continued support. We agree that the Adam optimizer's beta impacts the smoothness of the training curve and will prioritize further attention into this matter beyond the rebuttal period.

---

### Author Response · Authors · 2024-11-19
**Global Response**

We sincerely thank the chairs and reviewers for your voluntary efforts and valuable feedbacks. Here, we would like to we reiterate our contributions for your further consideration.

Our work contributes both practical value and theoretical significance to the community, summarized as follows:

- We are the first to identify Adam as the root cause of loss spikes during LLM training, as it can occasionally result in excessively large parameter updates (*Theorem 1*). To address this, we developed S3, which minimizes the risk of loss spikes by using the same coefficient for numerator and denominator momentum (*Theorem 2*). Prior to our work, practitioners had to rely on inefficient engineering solutions, such as skipping problematic data batches or restarting training from nearby checkpoints. These methods wasted significant time and hardware resources. As emphasized by **Reviewer Xi7Z** and **Reviewer Y3k2**, our work offers immediate practical utility and further valuable insights for the community.

- We challenge the prevailing belief that Adam’s effectiveness stems from its individually adaptive learning rates for each parameter. Instead, we identify that its efficacy primarily arises from its sign-like descent. This insight led to the development of S3, which features a generalized sign-like formulation with flexible $p$-order momentum, surpassing the constraints of fixed second-order momentum. Notably, this design ethos provides a foundation for constructing more advanced adaptive optimizers. This contribution was valued by **Reviewer grTo**.

- We incorporate the Nesterov Acceleration Gradient (NAG) technique into S3 to enhance training speed. While we are not the first to introduce NAG to adaptive optimizers, we derive a novel equivalence of NAG that eliminates additional memory costs and hyperparameter tuning (*Theorem 3*). This improvement is critical for training LLMs in practical settings.

- We introduce a novel theoretical framework to analyze the convergence rate of S3 under a weaker assumption of general non-uniform smoothness conditions (*Theorem 4, Remark 1*). This approach provides a fresh perspective for analyzing Adam and other adaptive optimizers, offering significant theoretical contributions to the community.  The theoretical contribution were acknowledged by **Reviewer cGUg ** and **Reviewer RDxB**.

-  We conducted extensive experiments to validate the causes of loss spikes and demonstrate the effectiveness of S3. **All reviewers** appreciated the soundness of our experimental results.

---

### Meta-Review · Area_Chair_c5ze · 2024-12-20

**Metareview:**

This paper investigates the effectiveness of Adam in training deep neural networks (DNNs), attributing its success to a sign-like approach. The paper also provides a theoretical analysis of the loss spikes encountered when using Adam for training large language models (LLMs), attributing these spikes to potentially large parameter updates. To address this, the authors propose a novel optimizer, SoftSignSGD (S3), which is claimed to offer multiple advantages over Adam. The work is supported by theoretical analysis and extensive experiments evaluating S3's performance.

Overall, while this work shows promise, it lacks sufficient depth and clarity in several areas.

1) Theoretical Limitations. A) The proposed theorem assumes that all gradients along the optimization trajectory share the same sign, which is highly restrictive and undermines the paper's motivation. B) The theoretical analysis only provides an upper bound on the update size, suggesting the potential for overly large updates. However, the absence of a lower bound leaves the claim of "large updates" as a hypothesis rather than a substantiated fact. The lower bound, which would guarantee a minimum scale for the update, is missing.

2) NAG and Higher-Order Moments. A) The equivalence formulations of Nesterov Accelerated Gradient (NAG) have already been proven in prior work, limiting the novelty of this contribution. B) Several existing works have integrated NAG into Adam-like optimizers (as cited in the paper), raising questions about the incremental insights provided here. C) The work does not clearly address any specific challenges in integrating NAG or higher-order moments, further limiting its novelty and impact.

Considering the current acceptance rate and the limitations in the paper's theoretical and practical contributions, we cannot recommend acceptance at this time. We encourage the authors to address these concerns thoroughly, as outlined in the reviewers' feedback, to strengthen the work for future submissions.

**Additional Comments On Reviewer Discussion:**

Here I mainly list the unresolved concerns from Reviewer grTo

(1)	use of strong assumption
The authors explain it but also acknowledge that the assumption may not hold.

(2)	 vacuousness of p-th order flexibility,
        The authors provide some explanations, but they are not so convincing.

(3)	 limited novelty on NAG,
The authors provide some explanations, but they are not so convincing.

(4)	statistical unreliability (only single run)
The authors provide some explanation: GPU limitations and high training cost.

My points are summarized below.
For (1), the authors require all gradients along the optimization trajectory to share the same sign, which is really restrictive. This directly challenges the motivation of this work, since the authors' claims are based on this theory. Moreover, the theory only provides the upper rather than the update's lower bound. The upper bound means that the update may be too large. The lower bound can show the update is at least at the scale, which is unfortunately missing. So the claim of large update is only a hypothesis instead of a fact.

For (2), extending the second-order moment to a flexible moment is not novel and not challenging for two reasons. 1) Padam already introduces a hyper-parameter p to the second moment vt as vtp, and thus have some similarity with  this work. 2) This work does not solve any extra challenges to apply for the higher-order moment. 3) The work does not provide any evidence to show the benefits of using higher-order moments, especially for theoretical aspects.

Padam: https://par.nsf.gov/servlets/purl/10214967

For (3) which introduces NAG, it is not novel for two reasons. Firstly, the equivalence formulations in theory are proved in previous papers instead of this work. Secondly, several works already integrated NAG into Adam-alike optimizers. See the citations in the paper. It does not see any challenges that this work resolves for NAG integration, limiting its insights and novelty.

For (4) statistical unreliability (only single run), I can understand the authors, since running an AI model, e.g., ViT-B and GPT2, multiple times is time-consuming and even challenging when the GPU resource is limited.

Overall, I doubt the insights and novelty of this work

---

### Decision · Program_Chairs · 2025-01-22

Reject